# Weak-to-Strong Generalization via Bregman Bias–Variance Decomposition

**Gengze Xu** [* 1 2 3] **Wei Yao** [* 1 2 3] **Ziqiao Wang** [† 4] **Yong Liu** [† 1 2 3]

## Abstract

Weak-to-strong generalization (W2SG) is the phenomenon in which a powerful student model, trained on labels produced by a weaker teacher, ultimately outperforms the teacher on the target task. In this work, we theoretically investigate how W2SG can arise via a generalized bias–variance decomposition under Bregman divergence. We show that the expected population risk gap between the student and the teacher is characterized by the expected misfit between the two models. Unlike earlier misfit-based analyses, our theory removes several restrictive assumptions, e.g., it does not require the student hypothesis class to be convex. Our results indicate that W2SG is more likely when the student effectively approximates the teacher's posterior mean. Specializing to squared loss, we provide a sufficient condition (illustrated through a concrete example) under which the student converges to its posterior mean teacher; in particular, increasing the student model size can ensure this convergence. For cross-entropy loss, our analysis further suggests that lowering the entropy of the student's predictive distribution can promote W2SG. We also find that the reverse cross-entropy, unlike the standard forward cross-entropy, is less sensitive to the teacher's predictive uncertainty. Finally, we verify these theoretical insights empirically and demonstrate that incorporating reverse cross-entropy consistently improves student performance.

## 1. Introduction

Superalignment for large language models (LLMs) is emerging as a fundamental challenge in ensuring the long-term safety of modern natural language processing (NLP) systems (OpenAI, 2023). A promising approach to addressing superalignment is weak-to-strong generalization (W2SG) (Burns et al., 2024), where a low-complexity weak teacher model generates pseudo-labels to supervise a high-complexity pre-trained strong student model, enabling the student to outperform the teacher on the target task. To theoretically understand the emergence of this phenomenon, recent advances in W2SG research propose several analytical frameworks (Ildiz et al., 2025; Wu & Sahai, 2025; Somerstep et al., 2025a; Dong et al., 2025; Charikar et al., 2024; Mulgund & Pabbaraju, 2025; Yao et al., 2025b;a; Lang et al., 2024; Shin et al., 2025; Gan et al., 2026). Among these, misfit-based analyses, introduced by Charikar et al. (2024), provide a key insight by linking the student's performance gain directly to the "misfit error", namely the discrepancy between the student's predictions and the labels provided by the teacher. While the original misfit analysis in Charikar et al. (2024) focuses on squared loss, Mulgund & Pabbaraju (2025) extends this analysis to a broader family of Bregman divergences. At a high level, these frameworks characterize the W2SG phenomenon through an inequality of the form:

$$\text{Error}_{\text{student}} \leq \text{Error}_{\text{teacher}} - (\text{Misfit} - \epsilon). \quad (\bigstar)$$

This inequality suggests that the student can outperform the teacher when the misfit term dominates the residual $\epsilon$, i.e., when $\epsilon$ is sufficiently small. However, previous misfit-based analyzes have several limitations. Firstly, they rely on convexity assumptions on the student function class (or convex-combination relaxations thereof), which are typically violated in modern deep classification setting, where softmax normalization renders the hypothesis class non-convex. Secondly, the bounds leave the residual term $\epsilon$ largely implicit, limiting interpretability and obscuring when it can vanish and thus when W2SG is guaranteed. In addition, these analyses typically ignore the statistical dependence between teacher and student induced by training procedures, and they often invoke realizability-style assumptions (e.g., that the ground-truth labeling function lies in the student hypothesis class) without explaining why a larger student is usually needed to observe W2SG. As a result, existing misfit-based

---

[*]Equal contribution [1]Gaoling School of Artificial Intelligence, Renmin University of China, Beijing, China [2]Beijing Key Laboratory of Research on Large Models and Intelligent Governance [3]Engineering Research Center of Next-Generation Intelligent Search and Recommendation, MOE [4]School of Computer Science and Technology, Tongji University, Shanghai, China. Correspondence to: Ziqiao Wang <ziqiaowang@tongji.edu.cn>, Yong Liu <liuyonggsai@ruc.edu.cn>.

*Proceedings of the 43rd International Conference on Machine Learning*, Seoul, South Korea. PMLR 306, 2026. Copyright 2026 by the author(s).

inequalities provide limited quantitative insight into sufficient conditions for W2SG, e.g., *how increasing student capacity affects the phenomenon*. Finally, previous results give little actionable guidance for practical algorithm design aimed at promoting W2SG, especially in the common setting where cross-entropy is used as the training loss.

In this work, we present new misfit-based W2SG inequalities by applying a generalized bias-variance decomposition of Bregman divergences (Pfau, 2013; Adlam et al., 2022), rather than relying on the generalized Pythagorean theorem used in (Charikar et al., 2024; Mulgund & Pabbaraju, 2025). Our novel proof strategy entirely removes both the convexity and realizability assumptions on the student hypothesis class, which clearly broadens the scope of misfit-based analyses and opens the door to more general results. In particular, while our inequalities retain the high-level structure of (★), our analysis is conducted at the level of *expected population risk*, taking expectations jointly over the data and model parameters. This joint viewpoint allows us to explicitly exploit the statistical dependence between teacher and student—an aspect not captured in earlier misfit analyses—and, crucially, obtains an analytic characterization of the residual term $\epsilon$. In fact, studying expected risk with respect to the randomness of model parameters is also standard in classical machine learning (Bishop, 2006, Section 3.2). This perspective further identifies an ideal scenario for W2SG occurs: when the student's predictions align with those of its *"posterior mean" teacher* (where the posterior is defined by conditioning the teacher's distribution on the student model), the residual $\epsilon$ vanishes. At a high level, this suggests using teacher model ensembles as a finite-sample proxy for the posterior mean teacher, which is consistent with previous empirical studies (Agrawal et al., 2025; Sang et al., 2024). However, the posterior mean teacher itself is generally not directly computable in practice. Therefore, it remains important to instantiate this sufficient condition in concrete settings.

To understand when the student approaches its posterior mean teacher, we first specialize to squared loss and show that reducing the expected misfit is sufficient to drive $\epsilon$ to zero, thus guaranteeing W2SG. We then address the remaining question of when the expected misfit can be small yet nonzero. In a concrete overparameterized ridge regression example, we prove that the expected misfit converges to a positive constant while $\epsilon$ vanishes once the student is sufficiently large, implying that the student converges in expectation to its posterior mean teacher. Notably, even though this ridge regression setting satisfies the convexity assumption of earlier work, the resulting quantitative insight on the role of student model size does not emerge from existing misfit-based analyses.

Beyond squared loss, our inequalities also provide concrete implications for cross-entropy (CE) and reverse cross-entropy (RCE). Specifically, our theory indicates that lower-entropy student predictions are more favorable for W2SG, and further shows that RCE is less sensitive to the teacher's predictive uncertainty than CE, an advantage when the teacher provides ambiguous supervision.

**Takeaways.** The key takeaways of our paper are summarized as follows:

- **Theoretical sufficient conditions for W2SG.** Our misfit-based W2SG inequalities, which require neither convexity nor realizability, identify student convergence to its posterior mean teacher as a sufficient condition for W2SG. In particular, in an overparameterized ridge regression example, we show that increasing student capacity can guarantee this convergence.

- **Implications for practical W2S training.** Lower predictive entropy in the student model strengthens W2SG under both CE and RCE. Moreover, RCE is significantly more robust than CE when the teacher exhibits high predictive uncertainty.

- **Empirical verification and algorithmic gains.** Our experiments corroborate the theoretical predictions. Furthermore, combining CE and RCE consistently improves W2SG in practice by more effectively utilizing teacher confidence.

## 2. Related Work

To address the challenge of superalignment (OpenAI, 2023), a growing body of research has built upon Burns et al. (2024), which empirically investigates the properties of W2SG (Shin et al., 2025; Yang et al., 2025; Goel et al., 2025; Yao et al., 2025b), and the potential of this paradigm on other tasks (Guo et al., 2024; Yang et al., 2024b) or scenarios (Pawelczyk et al., 2024; Zhou et al., 2025). Additionally, various techniques are also developed to enhance the strong model's performance in W2SG (Somerstep et al., 2025b). Popular approaches include iterative updating (Lyu et al., 2025; Ye et al., 2025; Lang et al., 2025), and incorporating more weak supervisors (Agrawal et al., 2025; Sang et al., 2024; Liu & Alahi, 2024; Cui et al., 2025).

In parallel, theoretical understanding of W2SG mainly focuses on whether it occurs, i.e. under what circumstances the strong student outperforms the weak teacher. Several works (Charikar et al., 2024; Mulgund & Pabbaraju, 2025; Yao et al., 2025b;a) analyze W2SG through projection-style geometric arguments, deriving risk bounds akin to the Pythagorean theorem that quantify how much a strong student model can outperform its weak teacher via their misfit error. Moreover, from the perspective of a general definition of adversarial robustness, W2SG arises under appropriate data neighborhood conditions that enable weak

supervision error correction (Lang et al., 2024) or sufficient overlap between easy and hard patterns that allow weak supervision to guide the student in learning challenging features (Shin et al., 2025). Under Gaussian data assumptions, the theoretical foundations of W2SG are rigorously characterized through several frameworks: model and distribution shift (Ildiz et al., 2025), transfer learning (Somerstep et al., 2025a) and intrinsic dimension (Dong et al., 2025). Further theoretical insights are established through representation analysis (Xue et al., 2025), feature learning (Wu & Sahai, 2025; Oh et al., 2025; Moniri & Hassani, 2025) and random feature model (Medvedev et al., 2025).

## 3. Preliminaries

**Notations.** Throughout this paper, unless otherwise stated, capital letters (e.g., $X$) denote random variables, while the corresponding lowercase letters (e.g., $x$) denote their realizations. Let $P_X$ be the distribution of $X$ and $P_{X|Y}$ be the conditional distribution of $X$ given $Y$. Conditioning on a specific realization is denoted by $P_{X|Y=y}$ or simply $P_{X|y}$. Expectations are expressed as $\mathbb{E}_X[\cdot]$. Similarly, $\mathbb{E}_{X|y}[\cdot]$ or $\mathbb{E}_X[\cdot \mid y]$ denotes expectation over $X \sim P_{X|Y=y}$.

**W2SG Problem Setup.** Let $\mathcal{X}$ and $\mathcal{Y}$ denote the instance and label domains, respectively. In the weak-to-strong setting, we consider two hypotheses classes $\mathcal{W} \subseteq \mathbb{R}^{d_w}$ and $\mathcal{W}' \subseteq \mathbb{R}^{d_s}$, with dimensions satisfying $d_s \geq d_w$. These hypothesis classes induce corresponding function classes: the weak model class $\mathcal{F} = \{f_w : \mathcal{X} \to \mathcal{Y} \mid w \in \mathcal{W}\}$, and the strong model class $\mathcal{F}' = \{f_{w'} : \mathcal{X} \to \mathcal{Y} \mid w' \in \mathcal{W}'\}$. We assume the existence of a ground-truth labeling function $g : \mathcal{X} \to \mathcal{Y}$. Suppose we have a pre-trained, high-capacity strong model $f_{w_0'} \in \mathcal{F}'$ and a weak model $f_w \in \mathcal{F}$, which may or may not be pre-trained. The weak model is trained on a dataset $S = \{(X_j, Y_j)\}_{j=1}^m$, drawn i.i.d. from an unknown distribution $\mu$. To leverage the weak model's supervision, we generate a pseudo-labeled dataset $S' = \{(X_i', Y_i')\}_{i=1}^n$, where $\{X_i'\}_{i=1}^n$ are drawn i.i.d. from $\mu_X$ (which is the marginal distribution of $X$ induced by $\mu$) and each pseudo-label is obtained as $Y_i' = f_w(X_i')$. Using this dataset, we fine-tune the pre-trained strong model $f_{w_0'}$ to obtain a final strong model $f_{w'}$. We say that weak-to-strong generalization occurs if this final strong model $f_{w'}$ estimates the ground-truth labeling function $g$ better than its weak teacher $f_w$.

**Remark 3.1** (Posterior Distribution of $f_W$). *The joint distribution of the teacher and student models, $P_{W,W'}$, is determined by: the marginal distribution of the teacher, $P_W$, which depends on the data distribution and other training randomness; and the conditional distribution $P_{W'|W}$, which captures the student's training given the teacher, including the influence of $S'$ and additional randomness such as pre-trained parameters $W_0$. Consequently, the conditional distribution $P_{W|W'}$ is well-defined and referred to as the "posterior" distribution of the teacher.*

**Background on Bregman Divergence.** To formally quantify how well a model estimates the ground-truth labeling function, a suitable loss function is typically employed. We now introduce the Bregman divergence (Bregman, 1967; Banerjee et al., 2005b;a), a general class of divergences containing many popular loss functions.

**Definition 3.1** (Bregman divergence). Let $\phi : \mathbb{R}^d \to \mathbb{R}$ be a differentiable, strictly convex function. The Bregman divergence generated by $\phi$ of $x, y \in \mathbb{R}^d$ is defined as $D_\phi(x, y) \triangleq \phi(x) - \phi(y) - \langle \nabla\phi(y), x - y \rangle$.

Common examples of Bregman divergences include the squared Mahalanobis distance, obtained by setting $\phi(x) = x^T M x$ for a positive semidefinite matrix $M \in \mathbb{R}^{d \times d}$ (which reduces to the squared loss or mean squared error if $M$ is the identity matrix, i.e. $\phi(x) = \|x\|^2$) and the Kullback–Leibler (KL) divergence, obtained by setting $\phi(x) = \sum_i^d x_i \log x_i$. Geometrically, the Bregman divergence measures the difference between the function $\phi$ and its supporting hyperplane at the point $y$. Although it generalizes the notion of a distance, sharing some metric-like properties such as non-negativity ($\forall x, y, D_\phi(x, y) \geq 0$) and the identity of indiscernibles ($\forall x, y, D_\phi(x, y) = 0$ if and only if $x = y$), it typically lacks symmetry and does not necessarily satisfy the triangle inequality. Instead, Bregman divergences obey a generalized law of cosines (Chen & Teboulle, 1993).

Let $\phi^*$ be the convex conjugate[1] of $\phi$, and $x^* = \nabla\phi(x)$ be the dual representation of $x$ under $\phi$ (we have $x = (x^*)^* = \nabla\phi^*(\nabla\phi(x))$ by properties of convex conjugation). Then, the generalized law of cosines for the Bregman divergence states that, for any $x, y, z \in \mathbb{R}^d$,

$$D_\phi(x, z) = D_\phi(x, y) + D_\phi(y, z) - \langle z^* - y^*, x - y \rangle. \quad (1)$$

Based on these properties, we formally define the dual expectation as follows.

**Definition 3.2** (Dual Expectation (Adlam et al., 2022)). The *dual expectation* of a random variable $X \in \mathbb{R}^d$ is defined as

$$\mathcal{E}[X] \triangleq (\mathbb{E}[X^*])^* = \nabla\phi^*(\mathbb{E}[\nabla\phi(X)]).$$

Below are two important minimization properties of Bregman divergences derived from Eq. (1).

**Lemma 3.1** ((Pfau, 2013; Adlam et al., 2022)). *Let $X$ be a random variable over $\mathbb{R}^d$, and $D_\phi$ be any Bregman divergence over $\mathbb{R}^d \times \mathbb{R}^d$. The minimizers of the expected Bregman divergence satisfy: (i)* $\arg\min_{y \in \mathbb{R}^d} \mathbb{E}[D_\phi(X, y)] = \mathbb{E}[X]$*, (ii)* $\arg\min_{y \in \mathbb{R}^d} \mathbb{E}[D_\phi(y, X)] = \mathcal{E}[X]$*.*

---

[1]For a function $\phi : \mathcal{X} \to \mathbb{R} \cup \{-\infty, +\infty\}$, its convex conjugate is $\phi^*(y) \triangleq \sup_{x \in \text{dom}(\phi)} \langle x, y \rangle - \phi(x)$.

Following (Pfau, 2013; Adlam et al., 2022), Lemma 3.1 allows us to derive the bias-variance decompositions of Bregman divergence w.r.t. an arbitrary point $y$,

$$\mathbb{E}\left[D_\phi\left(X, y\right)\right] = \mathbb{E}\left[D_\phi\left(X, \mathbb{E}\left[X\right]\right)\right] + D_\phi\left(\mathbb{E}\left[X\right], y\right), \quad (2)$$

$$\mathbb{E}\left[D_\phi\left(y, X\right)\right] = D_\phi\left(y, \mathcal{E}\left[X\right]\right) + \mathbb{E}\left[D_\phi\left(\mathcal{E}\left[X\right], X\right)\right]. \quad (3)$$

Given a Bregman divergence as the loss function, we define the *expected population risk* of the teacher model $f_W$ as $\mathbb{E}_{X,W}\left[D_\phi\left(g(X), f_W(X)\right)\right]$, where $X$ denotes an independent test sample. Since the Bregman divergence is generally asymmetric, we also define the corresponding *reverse population risk* as $\mathbb{E}_{X,W}\left[D_\phi\left(f_W(X), g(X)\right)\right]$. The population risks for the student model are defined analogously. Both the population risk and reverse population risk serve as measures of how well a model approximates the ground-truth labeling function $g$.

**Previous Misfit-based W2SG Result.** We next restate the previous misfit-based W2SG inequality from Mulgund & Pabbaraju (2025).

**Theorem 3.1** ((Mulgund & Pabbaraju, 2025, Thm 4.1), Informal[2]). *Let $f_{w'}$ and $f_w$ be the strong and weak models, obtained by adding a finetuning classification head on top of their respective pre-trained backbones. The student's classifier is drawn from a **convex set** $\mathcal{F}$. If $f_{w'}$ is sufficiently close to the projection of the weak model onto the strong model class, then*

$$\mathbb{E}_X\left[D_\phi\left(g(X), f_{w'}(X)\right)\right] \leq \mathbb{E}_X\left[D_\phi\left(g(X), f_w(X)\right)\right] - \mathbb{E}_X\left[D_\phi\left(f_{w'}(X), f_w(X)\right)\right] + \epsilon.$$

*The residual error $\epsilon$ vanishes when the student model $f_{w'}$ is exactly the projection.*

Theorem 3.1 suggests that a student can outperform a weak teacher, as quantified by their misfit, when the student is close to the projection of the teacher onto a convex hypothesis class. Several important caveats are worth highlighting. First, the expectation in Theorem 3.1 is taken over $X$ only, and therefore does not exploit the statistical dependence between $W$ and $W'$. Second, the convexity assumption is essential to their proof because the main technical tool, namely the generalized Pythagorean inequality, relies on projecting the weak model onto a convex model class. This requirement is typically violated in classification settings, where the softmax normalization renders the hypothesis class non-convex (Mulgund & Pabbaraju, 2025). Third, the residual $\epsilon$ term in Theorem 3.1 lacks a tractable analytic form, which limits interpretability and hinders further insight into when this term can vanish. Finally, the bound in Theorem 3.1 does not readily translate into actionable guidance for algorithm design aimed at promoting W2SG in practice.

---

[2]The formal statement is in Appendix A.

# 4. W2SG through the Lens of Expected Misfit

We are now in a position to present our main results.

**Theorem 4.1.** *Let $\zeta_1 = \sqrt{\mathbb{E}\left\|g(X) - f_{W'}(X)\right\|^2}$ and $\zeta_2 = \sqrt{\mathbb{E}\left\|g^*(X) - f_{W'}^*(X)\right\|^2}$. The following inequalities hold,*

$$\mathbb{E}\left[D_\phi\left(g(X), f_{W'}(X)\right)\right] \leq \mathbb{E}\left[D_\phi\left(g(X), f_W(X)\right)\right] - \mathbb{E}\left[D_\phi\left(f_{W'}(X), f_W(X)\right)\right] + \epsilon_1, \quad (4)$$

$$\mathbb{E}\left[D_\phi\left(f_{W'}(X), g(X)\right)\right] \leq \mathbb{E}\left[D_\phi\left(f_W(X), g(X)\right)\right] - \mathbb{E}\left[D_\phi\left(f_W(X), f_{W'}(X)\right)\right] + \epsilon_2. \quad (5)$$

*where $\epsilon_1 = \zeta_1\sqrt{\mathbb{E}\left\|f_{W'}^*(X) - \mathbb{E}\left[f_W^*(X) \mid W', X\right]\right\|^2}$ and $\epsilon_2 = \zeta_2\sqrt{\mathbb{E}\left\|f_{W'}(X) - \mathbb{E}\left[f_W(X) \mid W', X\right]\right\|^2}$.*

Eq. (4) implies that the student outperforms the teacher (i.e., W2SG occurs) whenever $\epsilon_1$ is sufficiently small. Moreover, the potential performance gain is now characterized by the *expected reverse misfit*, namely $\mathbb{E}\left[D_\phi\left(f_{W'}(X), f_W(X)\right)\right]$, between the two models. A key difference from Theorem 3.1 is that our expectation[3] is taken jointly over both data and parameter randomness, i.e., under $P_{X,W,W'}$. This joint view is central to our analysis. In particular, it enables us to remove the convexity assumption (as well as other assumptions such as realizability and sequential consistency used in earlier works; see Theorem A.2 in Appendix), to capture the statistical dependence between the teacher and student, and, crucially, to derive an explicit expression for the residual $\epsilon_1$, a quantity left implicit in earlier misfit-based analyses. Technically, these improvements stem from invoking the Bregman bias-variance decomposition in Eq. (3), rather than projecting the teacher onto a convex student hypothesis class. Note that working with expected loss is the de facto convention in classical machine learning. For example, the classical bias–variance decomposition in Bishop (2006, Section 3.2) considers a fixed unseen data and takes expectation over the randomness in the learned parameters induced solely by the training dataset.

In addition, Eq. (5) conveys an analogous message, but focuses on reverse population risks, a direction of W2SG inequality not established by Mulgund & Pabbaraju (2025). In fact, the *expected forward misfit*, namely $\mathbb{E}\left[D_\phi\left(f_W(X), f_{W'}(X)\right)\right]$, in Eq. (5), becomes, when instantiated with cross-entropy (a surrogate for KL divergence), exactly the standard training objective used in practical W2SG setups. We elaborate on this in Section 6.

Notably, Theorem 4.1 reveals a subtle trade-off in aligning the student model with the teacher. On one hand, since the student lacks access to the ground-truth labels, it must

---

[3]To avoid clutter, we omit expectation subscripts in theorem statements.

rely on the pseudo labels provided by the teacher. That is, minimizing the empirical risk with respect to the teacher's outputs, e.g., $\frac{1}{n} \sum_{i=1}^{n} \mathrm{D}_\phi \left( f_w(x_i), f_{w'}(x_i) \right)$, becomes a necessary part of training. On the other hand, Eq. (4-5) suggest that the expected misfit between the teacher and the student models contributes directly to the performance improvement of the strong student over the weak teacher. In particular, greater misfit between the two models can indicate a larger performance gain achieved by the student, which also align with a recent argument given in Dong et al. (2025).

Theorem 4.1 also hints conditions under which $\epsilon_1$ and $\epsilon_2$ vanish, leading to the following results.

**Corollary 4.1.** *Under the teacher-student setting, if $f_{W'}(x) = \mathcal{E}\left[f_W(x)|W'\right]$ for $\forall x \in \mathcal{X}$, then*

$$
\mathbb{E}\left[\mathrm{D}_\phi\left(g(X), f_{W'}(X)\right)\right] = \mathbb{E}\left[\mathrm{D}_\phi\left(g(X), f_W(X)\right)\right] \\
- \mathbb{E}\left[\mathrm{D}_\phi\left(\mathcal{E}\left[f_W(X)|W', X\right], f_W(X)\right)\right]. \quad (6)
$$

*Furthermore, if $f_{W'}(x) = \mathbb{E}\left[f_W(x)|W'\right]$ for $\forall x \in \mathcal{X}$, then*

$$
\mathbb{E}\left[\mathrm{D}_\phi\left(f_{W'}(X), g(X)\right)\right] = \mathbb{E}\left[\mathrm{D}_\phi\left(f_W(X), g(X)\right)\right] \\
- \mathbb{E}\left[\mathrm{D}_\phi\left(f_W(X), \mathbb{E}\left[f_W(X)|W', X\right]\right)\right]. \quad (7)
$$

**Remark 4.1.** *Corollary 4.1 presents situations where W2SG is guaranteed to emerge. Specifically, when the student's prediction matches that of its (dual) "posterior mean" teacher, where the (dual) expectation is taken with respect to $P_{W|W'}$ (i.e. the posterior distribution of teacher model, as discussed in Remark 3.1). Clearly, the posterior mean teacher is generally not directly computable in practice. However, a proxy, averaging labels or predictions from multiple teachers provides a practical way to approximate this quantity. Therefore, we empirically evaluate this ensemble-based strategy in Section 7 as a sanity check of the theory.*

Strictly speaking, Corollary 4.1 by itself does not clarify when the residuals vanish, which motivates a closer look at the regime where the student approaches its posterior mean teacher. In addition, although the residual terms ($\epsilon_1$ and $\epsilon_2$) now admit explicit expressions, their relationship to the expected misfit is not yet well understood at this stage. In the next section, we show that decreasing the expected misfit provides a sufficient condition for driving the residuals to zero. We further demonstrate that, in overparameterized settings, increasing the student model size can indeed lead to convergence to its posterior mean teacher.

## 5. Symmetric Bregman: Squared Loss

In Mulgund & Pabbaraju (2025), the emergence of W2SG is attributed to the student being close to the convex projection of the teacher, but it is unclear why fine-tuning should lead to this projection. In contrast, our Theorem 4.1 suggests that W2SG arises when the student model approximates the

posterior mean teacher. We now investigate the conditions under which the student model becomes close to its posterior mean teacher, thereby driving $\epsilon_1$ and $\epsilon_2$ to zero.

Specifically, in this section, we focus on the squared loss (i.e. $\phi(x) = \|x\|^2$), where $\zeta_1 = \zeta_2$ and $\epsilon_1 = \epsilon_2$. The following result is a special case of Theorem 4.1.

**Corollary 5.1.** *Let $\phi(x) = \|x\|^2$, then the following inequality holds,*

$$
\mathbb{E}\|g(X) - f_{W'}(X)\|^2 \le \mathbb{E}\|g(X) - f_W(X)\|^2 \\
- \mathbb{E}\|f_{W'}(X) - f_W(X)\|^2 + \epsilon_2, \quad (8)
$$

*where $\epsilon_2 = 2\zeta_1 \sqrt{\mathbb{E}\left\|f_{W'}(X) - \mathbb{E}\left[f_W(X) \mid W', X\right]\right\|^2}$.*

To analyze when $\epsilon_2$ becomes small, we focus on the quantity $\mathbb{E}\left\|f_{W'}(X) - \mathbb{E}\left[f_W(X) \mid W', X\right]\right\|^2$, rather than directly analyzing the expected population risk of student model $\mathbb{E}\left\|g(X) - f_{W'}(X)\right\|^2$, i.e. the $\zeta_1^2$ term. For any $x \in \mathcal{X}$,

$$
\mathbb{E}\left\|f_{W'}(x) - \mathbb{E}\left[f_W(x) \mid W'\right]\right\|^2 = \underbrace{\mathbb{E}\left\|f_{W'}(x) - f_W(x)\right\|^2}_{\text{Expected Misfit}} \\
- \underbrace{\mathbb{E}\left\|f_W(x) - \mathbb{E}\left[f_W(x) \mid W'\right]\right\|^2}_{\text{Conditional Variance of } f_W}. \quad (9)
$$

Notably, due to the presence of the conditional variance term, we can see that $\epsilon_2$ vanishes before the expected misfit does, in other words, the vanishing of the expected misfit term is a sufficient condition for $\epsilon_2$ to vanish. Furthermore, recent studies on the "double descent" phenomenon show that this expected misfit, when regarded as the population risk, can indeed decrease as the capacity of the student model increases (Belkin et al., 2019; Hastie et al., 2022; Mei & Montanari, 2022; Yang et al., 2020; Ba et al., 2020). We now formalize this through the following example.

**Example 1** (Ridge Regression). Assume $X \sim \mathcal{N}\left(0, \mathbf{I}_{d_w}/d_w\right)$, and let the teacher model be $f_w(x) = x^\top w$ for $w \in \mathbb{R}^{d_w}$ and the student model be $f_{w'}(x) = (w'_1 x)^\top w'_2$, where[4] $w'_1 \in \mathbb{R}^{d_s \times d_w}$ and $w'_2 \in \mathbb{R}^{d_s}$. The weights $W'_1$ are initialized with entries drawn i.i.d. from $\mathcal{N}(0, 1/d_w)$ and remain fixed during training. The student is trained via ridge regression: $\min_{w'_2} \|(w'_1 \mathbf{x}')^\top w'_2 - \mathbf{y}'\|^2 + \eta\|w'_2\|^2$ where $\mathbf{x}' = [x'_1, \ldots, x'_n] \in \mathbb{R}^{d_w \times n}$ and $\mathbf{y}' = [y'_1, \ldots, y'_n] \in \mathbb{R}^n$.

Consider an overparameterized setting, we have the following asymptotic result.

**Theorem 5.1.** *Assume $W \sim \mathcal{N}(\mathbf{m}, B\mathbf{I}_{d_w})$ where $\mathbf{m} \in \mathbb{R}^{d_w}$ is independent of $W'_1$ and $\mathbf{X}'$, and $\|\mathbf{m}\|^2 \le c$ for some fixed constant c. Suppose that $\frac{d_s}{d_w} = \gamma_1 \to \infty$ and*

---

[4] Here $d_s$ refers to the number of hidden units rather than the total number of parameters.

$\frac{n}{d_w} \to \gamma_2 \in (0,1)$ *as* $n, d_w, d_s \to \infty$. *Then, for fixed* $\eta > 0$, *as* $n, d_s, d_w \to \infty$, *the trained ridge student satisfies*

$$\mathbb{E}\|f_{W'}(X) - f_W(X)\|^2 \to B(1 - \gamma_2).$$

Theorem 5.1 shows that, in the overparameterized regime $n < d_w$, the expected misfit term converges to the nonzero limit $B(1 - \gamma_2)$ as $\gamma_1 \to \infty$, that is, when the student is asymptotically much larger than the teacher. In fact, in this asymptotically isotropic setting, the misfit term is monotonically decreasing in $\gamma_1$ up to an $o(1)$ error. In other words, increasing the student capacity asymptotically reduces the misfit.

The following corollary further demonstrates that, in this setting, the misfit term remains nonzero while $\epsilon_2$ vanishes, thus guaranteeing the emergence of W2SG.

**Corollary 5.2.** *Under the same conditions in Theorem 5.1, we have* $\epsilon_2 \to 0$.

**Remark 5.1.** *Corollary 5.2 provides a concrete mechanism for the emergence of W2SG in our setting. When the student model is sufficiently large, it converges in expectation to the posterior mean teacher. Equivalently, the RHS of Eq. (9) vanishes, and hence $\epsilon_2 \to 0$ in Eq. (8). Meanwhile, the expected misfit between the student and an individual teacher remains strictly positive, namely $\mathbb{E}\|f_{W'}(X) - f_W(X)\|^2 \to B(1 - \gamma_2) > 0$. Thus, the student approaches the posterior mean teacher rather than merely imitating a particular teacher realization. Since the expected performance gain is characterized by the difference between the misfit term and $\epsilon_2$, the vanishing of $\epsilon_2$ together with a nonzero misfit obtains a strictly positive gain, and hence W2SG emerges. It is also worth emphasizing that this result suggests that enlarging the student makes W2SG easier to observe, which is the focus of this section. However, it does not imply that increasing the student capacity necessarily maximizes the achievable performance gain.*

Example 1 and Theorem 5.1 can be extended to nonlinear neural networks by utilizing the analysis in Mei & Montanari (2022). In summary, W2SG is more likely to arise when the student model is sufficiently large (i.e., large $\gamma_1$), but the performance gain eventually saturates; as the student size grows, the performance gain converges to its minimum value (e.g., $B(1 - \gamma_2)$ in Example 1). Note that even if we assume the student hypothesis class is convex, which indeed holds in the random-feature–style regression setting of Example 1, these quantitative insights are still absent from previous misfit-based analyses (Charikar et al., 2024; Mulgund & Pabbaraju, 2025). Existing results typically rely on a realizability-type assumption that the ground-truth labeling function lies in the student's hypothesis class, without clarifying why increased student capacity is needed or how enlarging the student affects the emergence of W2SG.

While we show that enlarging the student model enables convergence to its posterior mean teacher under a symmetric Bregman divergence (i.e. squared loss), we conjecture that a similar result should extend to asymmetric Bregman divergences. Establishing this theoretically, however, remains challenging. A key obstacle is the absence of formal analyses of double descent under cross-entropy loss, despite its well-known empirical evidence (Nakkiran et al., 2020).

## 6. Asymmetric Bregman: From KL to CE

Although providing a formal justification for the vanishing of $\epsilon_1$ or $\epsilon_2$ under asymmetric Bregman divergences is difficult, our two-direction misfit-based W2SG inequalities in Theorem 4.1 nevertheless provide valuable insights into this setting.

We now turn to a $K$-class classification task, where $\mathcal{Y} \subset \mathbb{R}^K$ and $\|y\|_1 = 1$ for all $y \in \mathcal{Y}$ (i.e. $\mathcal{Y}$ is a simplex), and consider the commonly used CE loss function (as in the original W2SG paper (Burns et al., 2024)), defined as $\mathrm{CE}(y, \hat{y}) \triangleq -\sum_{i=1}^K y_i \log \hat{y}_i$ for any $y, \hat{y} \in \mathcal{Y}$. Since $\mathrm{CE}(y, \hat{y}) = \mathrm{D_{KL}}(y\|\hat{y}) + H(y)$, where $\mathrm{D_{KL}}(y\|\hat{y}) \triangleq \sum_{i=1}^K y_i \log \frac{y_i}{\hat{y}_i}$ is the KL divergence and $H(y) = -\sum_{i=1}^K y_i \log y_i$ is the Shannon entropy, and given that KL divergence is a special case of Bregman divergence, we can derive the following result from Theorem 4.1.

**Corollary 6.1.** *Let* $\phi(x) = \sum x_i \log x_i$ *and define the reverse cross-entropy (RCE) as* $\mathrm{RCE}(y, \hat{y}) \triangleq -\sum_{i=1}^K \hat{y}_i \log y_i$ *for any* $y, \hat{y} \in \mathcal{Y}$, *then*

$$\mathbb{E}[\mathrm{CE}(g(X), f_{W'}(X))] \leq \mathbb{E}[\mathrm{CE}(g(X), f_W(X))] - \mathbb{E}[\mathrm{RCE}(f_W(X), f_{W'}(X))] + \mathbb{E}[H(f_{W'}(X))] + \epsilon_1,$$

$$\mathbb{E}[\mathrm{RCE}(g(X), f_{W'}(X))] \leq \mathbb{E}[\mathrm{RCE}(g(X), f_W(X))] - \mathbb{E}[\mathrm{CE}(f_W(X), f_{W'}(X))] + \mathbb{E}[H(f_{W'}(X))] + \epsilon_2.$$

Notably, both inequalities in Corollary 6.1 involve the entropy of the student's prediction, $H(f_{W'}(X))$. This implies that, when CE or RCE is used to measure population risk, reducing the entropy of the student's output distribution favors the emergence of W2SG. In other words, high-confidence predictions by the student (i.e., low predictive uncertainty) are beneficial for outperforming the teacher. In fact, the original W2SG paper (Burns et al., 2024) adopts a regularized loss of the form $\mathcal{L}_{\mathrm{AUX}} = \beta \mathrm{CE}(f_w(x), f_{w'}(x)) + (1-\beta)\mathrm{CE}(\hat{f}_{w'}(x), f_{w'}(x))$, where $\hat{f}_{w'}(x)$ is the hardened student's prediction[5] so $\hat{f}_{w'}(x)$ is a one-hot vector. This regularization explicitly encourages entropy minimization, consistent with the implications of Corollary 6.1.

---

[5]In Burns et al. (2024, Eq. (1)), a confidence threshold is applied to decide whether to harden the prediction.

Furthermore, evaluating the quality of a trained model using the forward CE is the de facto standard in practice, then according to Corollary 6.1, minimizing RCE between the teacher and student appears more natural. As conjectured at the end of Section 5, this may help reduce $\epsilon_1$ and thereby facilitate W2SG. Empirical studies by Yao et al. (2025a) have also advocated reverse KL for its mode-seeking property, which benefits strong model performance. We now proceed to elaborate on the comparison between CE and RCE as objective functions in W2S training. In the idealized setting where both $\epsilon_1$ and $\epsilon_2$ vanish, we obtain the following result.

**Proposition 1.** *Under the ideal conditions of Corollary 4.1 where $\epsilon_1$ and $\epsilon_2$ vanish, the performance gain in the first inequality of Corollary 6.1 coincides with Eq. (6), while the gain in the second inequality is strictly smaller than Eq. (7).*

Thus, evaluating the population risk using CE, while aligning the student to the teacher via RCE, preserves the performance gains predicted by Corollary 4.1. Moreover, RCE provides additional notable advantages when the teacher's predictions exhibit low confidence (i.e., high uncertainty), as illustrated below.

**Proposition 2.** *Given a data distribution $\mu = \mu_X \mu_{Y|X}$ and any $\alpha \in [0, 1]$, we define a label-shifted distribution $\hat{\mu} = \mu_X \mu_{\hat{Y}|X}$ by smoothing the labels as follows: for each $(X, Y) \sim \mu$, the smoothed label is given by $\hat{Y}_j = \frac{1}{2} + \alpha \left( Y_j - \frac{1}{2} \right)$ for $\forall j \in [2]$. If RCE is used as the loss function in binary classification, then the population risk minimizer on $\hat{\mu}$ also minimizes the population risk on $\mu$.*

Note that decreasing $\alpha$ increases the uncertainty of the smoothed label $\hat{Y}$ but does not affect its hard label. If $\hat{Y}$ is the label provided by the teacher, Proposition 2 shows that RCE is less sensitive to the teacher's confidence levels. This property is especially desirable when the input $x$ is ambiguous for the teacher. Furthermore, in Appendix D.4, we demonstrate that uncertain weak supervision can lead to vanishing gradients under standard CE, whereas RCE maintains gradient stability. Later on we will empirically demonstrate how to use the advantages of RCE in scenarios involving low-confidence labels.

## 7. Experiments

In this section, we present empirical studies on W2SG. Specifically, we verify our theoretical insights, visualize the bias and variance terms in W2SG, and compare CE and RCE as training objectives. We also propose a novel training objective to improve the performance of W2SG. Our code for reproducing the experiments is available at https://github.com/Gengze/W2SG-Bregman.

**Datasets.** Our experiments employ diverse datasets covering both standard NLP tasks and LLM reward modeling tasks. For standard NLP tasks, we utilize the SciQ (Welbl et al., 2017), Amazon Polarity (McAuley & Leskovec, 2013) and Twitter Sentiment [6] datasets, following the experimental setup of Burns et al. (2024) to transform these datasets into binary classification tasks. For reward modeling tasks, we sample subsets from CAI-Harmless (Bai et al., 2022b) and HH-RLHF (Bai et al., 2022a), with experimental settings referencing Yang et al. (2025), to guide models toward achieving harmlessness or helpfulness objectives.

**Models.** We use models from the GPT-2 series (Radford et al., 2019), including GPT2, GPT2-Medium, GPT2-Large, and GPT2-XL, and the Qwen series (Bai et al., 2023), including Qwen-1.8B, Qwen-7B, and Qwen-14B. We use full fine-tuning without freezing any pretrained parameters.

**W2S Training Pipeline.** For each teacher-student pair, we use the standard two-stage training pipeline: (i) teacher training, where we train a weak model on data with ground-truth labels via supervised learning; and (ii) W2S training, where the weak model generates pseudo-labels for a new batch of data, which are then used to supervise a strong model. Unless otherwise specified, other settings remain the same as in the teacher training stage.

**Emergence of W2SG.** Guided by our theoretical analysis, we empirically investigate the mechanisms driving W2SG in two key aspects: the role of the teacher's posterior mean and the impact of the student's capacity. First, to verify the sufficient condition outlined in Corollary 4.1, we approximate the posterior mean teacher using a probability-based ensemble of multiple weak teachers (Dietterich, 2000; Zhou, 2025), investigating whether this averaged supervision suffices to trigger W2SG. Specifically, we construct multiple weak teachers by fine-tuning the same pretrained model on disjoint data subsets, and combine their predictions by taking the dual expectation as the proxy for the posterior mean teacher. Second, to validate the insights from Theorem 5.1 regarding model size, we progressively scale the strong model's capacity. We use Algorithm 1 (Yang et al., 2020) to estimate the changes in bias and variance. The results are shown in Figure 1 and Figure 3.

As shown in Figure 1, using model ensembles (dashed curves) consistently achieves lower test loss compared to supervision from a single teacher (solid curves). This improvement stems primarily from variance reduction, as incorporating more weak teachers helps mitigate the randomness in their outputs. These results demonstrate that when the student approaches its posterior mean teacher, its performance surpasses that of mimicking an individual teacher, indicating that W2SG is more likely to occur. This observation aligns with prior studies on ensemble learning and boosting

---

[6]Dataset available on Kaggle: https://www.kaggle.com/datasets/jp797498e/twitter-entity-sentiment-analysis

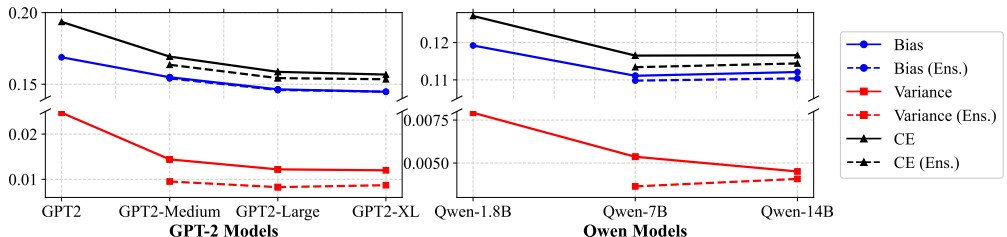

*Figure 1.* Bias-variance decomposition of the CE loss on Amazon Polarity. For each model family, we fix the weak teacher and evaluate students of increasing sizes: GPT2 supervises GPT2-Medium, GPT2-Large, and GPT2-XL, while Qwen-1.8B supervises Qwen-7B and Qwen-14B. The black curves show the CE loss, and the blue/red curves show its bias and variance components. Solid lines represent the mean performance of students supervised by individual weak teachers, while dashed lines denote performance under teacher ensemble supervision (Ens.).

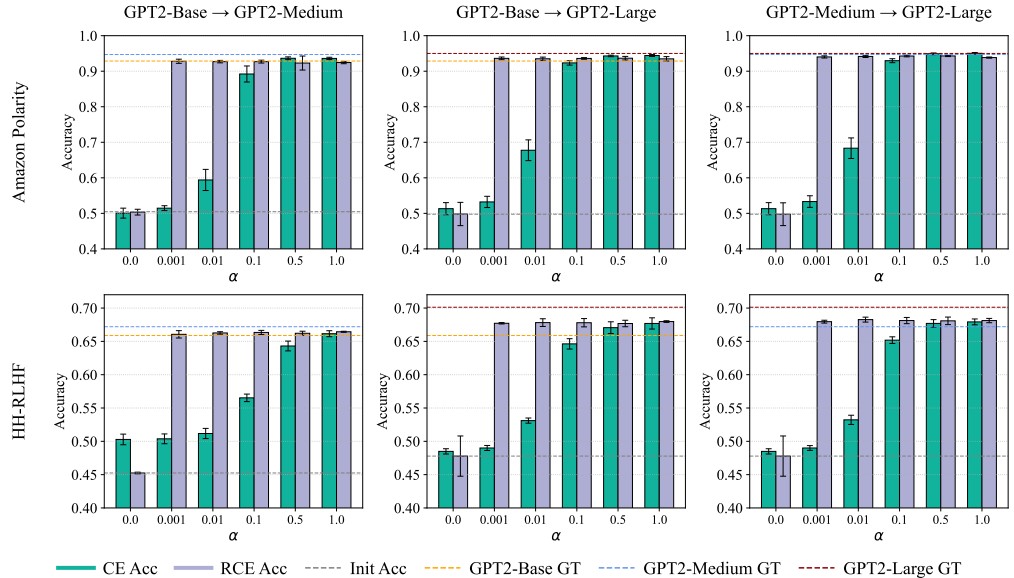

*Figure 2.* Comparison of CE and RCE losses. Lower $\alpha$ indicates higher uncertainty. "Weak $\rightarrow$ Strong" denotes the teacher-student pair. GT denotes training with ground truth labels. RCE demonstrates remarkable robustness to the teacher's prediction confidence.

in W2SG (Agrawal et al., 2025; Sang et al., 2024; Liu & Alahi, 2024; Cui et al., 2025), where multiple teachers are employed to train a student model.

Furthermore, scaling up the student's capacity drives it closer to the posterior mean teacher (the black solid and dashed lines moves closer), and primarily reduces the bias term, which is significantly larger than the variance term and dominates the risk. This finding differs from Dong et al. (2025), which mainly focuses on variance-dominated scenarios. However, our trends align with Chen et al. (2024), which shows that bias and variance often exhibit consistent trends, and the variance is upper bounded by the bias.

**Robustness of RCE.** To verify our theoretical claim that RCE is less sensitive to the teacher's prediction confidence, we compare CE and RCE under varying levels of supervision uncertainty using the label smoothing strategy defined in Proposition 2. As shown in Figure 2, RCE demon-

strates remarkable robustness, maintaining high accuracy even when the teacher's predictions are nearly random ($\alpha = 0.001$). In contrast, the performance of CE drops rapidly as $\alpha$ decreases. This aligns with our theoretical analysis in Proposition 2, which shows that highly uncertain supervision severely weakens the training signal under CE, while RCE preserves informative and stable gradients. To further validate this, we analyze the gradient dynamics of CE/RCE and extend our experiments to broader datasets, larger models, alternative loss functions (KL/Reverse KL), and general teacher–student settings. The results in Appendix E.3 consistently reinforce our conclusions.

**Combining CE and RCE.** Motivated by the observation that RCE excels in low-confidence scenarios while CE is effective in high-confidence regimes, we propose combining them to maximally exploit the teacher's signals. To explore this, we propose confidence-adaptive cross entropy (CACE)

*Table 1.* Test accuracy (%) of five loss functions. The optimal and suboptimal results are marked in **bold** and underline, respectively.

| Dataset | Model | CE | RCE | AUX | CACE | SL |
|---|---|---|---|---|---|---|
| CAI-Harmless | GPT2 → GPT2-Medium | 93.05 | 93.00 | 92.81 | **93.44** | 93.32 |
| | GPT2 → GPT2-Large | 93.88 | 94.55 | 94.51 | **94.78** | 94.63 |
| | GPT2-Medium → GPT2-Large | 95.47 | 95.40 | 95.40 | **95.62** | 95.58 |
| | Qwen-1.8B → Qwen-7B | 96.45 | 95.93 | 96.40 | **96.75** | 96.50 |
| | Qwen-1.8B → Qwen-14B | 96.03 | 94.93 | 95.65 | **96.08** | 95.70 |
| | Qwen-7B → Qwen-14B | 96.48 | 95.75 | 96.13 | **96.58** | 96.13 |
| HH-RLHF | GPT2 → GPT2-Medium | 66.15 | 66.44 | 66.21 | 66.18 | **66.65** |
| | GPT2 → GPT2-Large | 67.69 | **67.97** | 67.73 | 67.82 | **67.97** |
| | GPT2-Medium → GPT2-Large | 67.93 | 68.13 | 68.08 | 68.11 | **68.53** |
| | Qwen-1.8B → Qwen-7B | 68.45 | 69.08 | 68.80 | 69.20 | **69.25** |
| | Qwen-1.8B → Qwen-14B | 66.03 | 67.45 | 67.73 | **69.20** | 68.65 |
| | Qwen-7B → Qwen-14B | 67.90 | 69.13 | 69.20 | 69.78 | **70.85** |

loss as:

$$\mathcal{L}_{\text{CACE}}(y,\hat{y}) = \mathbb{I}(y,c)\cdot\text{RCE}(y,\hat{y})+\big(1-\mathbb{I}(y,c)\big)\cdot\text{CE}(y,\hat{y}),$$

where $c$ is the confidence threshold, and $\mathbb{I}(y,c)$ is an indicator function that activates when the confidence of the soft label $y$ is below $c$. We select $c$ by a quantile-based rule, with a hyperparameter $\eta\%$ specifying the percentage of samples treated as low-confidence. More details are provided in Appendix E.1.

We note that the symmetric cross entropy loss (SL) proposed by Wang et al. (2019) shares a similar design philosophy, defined as:

$$\mathcal{L}_{\text{SL}}(y,\hat{y}) = \lambda_1\text{RCE}(y,\hat{y}) + \lambda_2\text{CE}(y,\hat{y}),$$

where $\lambda_1$ and $\lambda_2$ are trade-off hyperparameters. Additionally, we compare our method with the auxiliary confidence loss (AUX) from Burns et al. (2024), corresponding to $\mathcal{L}_{\text{AUX}}$ introduced in Section 6. As shown in Table 1, CACE or SL consistently outperforms other three losses. While AUX improves student confidence via regularization, it requires careful hyperparameter tuning for the warm-up phase. In contrast, CACE and SL adapt purely based on the teacher's confidence, making them more robust and easier to tune. Additional results are provided in Table 3 in Appendix.

## 8. Conclusion and Limitations

This work provides a theoretical and empirical analysis of W2SG. We identify that a sufficient condition for W2SG is the student approximating its posterior mean teacher, which can be realized by increasing the student's model size. Furthermore, our results indicate that higher student prediction confidence facilitates W2SG, and demonstrate the effectiveness of RCE under uncertain supervision, providing guidance for practical W2S training. Some of our sharpest quantitative insights are obtained in the ridge regression setting. This focus reflects the current limits of deep learning

theory, particularly our incomplete understanding of double descent in fully non-convex models. Extending these analyses to broader settings is an important direction for future work. Moreover, while we establish sufficient conditions for W2SG, a more complete account of W2SG in practice will likely require understanding how pre-training shapes representations and inductive biases, an aspect that lacks quantitative analysis in existing W2SG theory. Finally, beyond the single-task setting, extending our Bregman bias–variance decomposition framework to semi-supervised multi-objective learning (Wegel et al., 2026) is also a promising direction.

## Acknowledgements

This research was supported by the National Key Research and Development Program of China (No. 2024YFE0203200), the National Natural Science Foundation of China (Nos. 62506265 and 62476277), and the Beijing Natural Science Foundation (No. Z250001), the CCF-ALIMAMA TECH Kangaroo Fund (No. CCF-ALIMAMA OF 2024008), and the Huawei–Renmin University joint program on Information Retrieval. We also acknowledge the support provided by the fund for building worldclass universities (disciplines) of Renmin University of China, and by the funds from the Beijing Key Laboratory of Big Data Management and Analysis Methods, Gaoling School of Artificial Intelligence, Renmin University of China, the Engineering Research Center of Next-Generation Intelligent Search and Recommendation, Ministry of Education, the Intelligent Social Governance Interdisciplinary Platform, Major Innovation & Planning Interdisciplinary Platform for the "DoubleFirst Class" Initiative, Renmin University of China, the Public Policy and Decision-making Research Lab of Renmin University of China, and the Public Computing Cloud, Renmin University of China.

## Impact Statement

This paper presents work whose goal is to advance the field of Machine Learning. There are many potential societal consequences of our work, none which we feel must be specifically highlighted here.

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

# Appendix

## A. Previous Misfit-based Results

**Theorem A.1** (Restatement of Theorem 1 from Charikar et al. (2024)). *This Theorem considers the squared loss as a special case of Bregman divergence $D_\phi(\cdot, \cdot)$, i.e., $\phi(x) = \|x\|^2$. Let $h_s : \mathbb{R}^d \to \mathbb{R}^{d_s}$ and $h_w : \mathbb{R}^d \to \mathbb{R}^{d_w}$ be the strong and weak model representation maps respectively. Given some data labeled by $g$, let $f_w \circ h_w$ be the function learnt by the weak model, for some classier head $f_w : \mathbb{R}^{d_w} \to \mathbb{R}$. For a convex set of functions $\mathcal{F}_s$ mapping $\mathbb{R}^{d_s}$ to $\mathbb{R}$ let*

$$f_{sw} = \operatorname{argmin}_{f \in \mathcal{F}_s} \mathbb{E}_X \|f(h_s(X)) - f_w(h_w(X))\|^2,$$

*be the function learnt by the strong model under weak supervision. Lastly, let us assume that there exists $f_s \in \mathcal{F}_s$ such that $f_s \circ h_s = g$. Then, we have that*

$$\mathbb{E}_X \|f_{sw}(h_s(X)) - g(X)\|^2 \leq \mathbb{E}_X \|f_w(h_w(X)) - g(X)\|^2 - \mathbb{E}_X \|f_{sw}(h_s(X)) - f_w(h_w(X))\|^2. \tag{10}$$

**Remark A.1.** *In comparison, our bound in Corollary 5.1 is*

$$\mathbb{E}_{X,W'} \|g(X) - f_{W'}(X)\|^2 \leq \mathbb{E}_{X,W'} \|g(X) - f_W(X)\|^2 - \mathbb{E}_{X,W',W} \|f_{W'}(X) - f_W(X)\|^2 + \epsilon_2,$$

*Our bound differs from Theorem A.1 in that it focuses on the expected population risk, that is, the expectation is taken over both the data distribution and the model parameters. As mentioned in the main text, this enables us to remove the convexity assumption by invoking Lemma 3.1. Notably, when restricted to convex function classes, our result recovers Theorem A.1, as convexity guarantees the last term $\mathbb{E}_{W'}\left\langle \mathbb{E}_{W|W'}\left[f_W^*(x)\right] - f_{W'}^*(x), g(x) - f_{W'}(x)\right\rangle \leq 0$ in Eq. (16) in our framework, following similar developments as those in Charikar et al. (2024). This recovery can also be seen by comparing (Charikar et al., 2024, Eq. (8)) with our Eq. (16). Specifically, taking expectations over both $(W, W') \sim P_{W,W'}$ in their bound obtains a result similar to ours, corresponding to a non-positive residual term in Eq. (16) in our proof.*

**Theorem A.2** (Restatement of Theorem 4.1 from (Mulgund & Pabbaraju, 2025)). *Let $\phi$ be a proper convex function as defined in Definition 3.1. Let $h_s$, $h_w$ be defined in the same way in Theorem A.1. Let $f_w : \mathbb{R}^{d_w} \to \mathcal{Y}$ be the weak model finetune layer, and $g : \mathcal{X} \to \mathcal{Y}$ be the target function. Let $\mathcal{F}$ be a class of functions mapping $\mathbb{R}^{d_s} \to \mathcal{Y}$. If the following hold:*

1. *(Realizability) $\exists f_* \in \mathcal{F}$ s.t. $g = f_* \circ h_s$,*

2. *(Convexity) $\mathcal{F}$ is a convex set of functions,*

3. *(Sequential Consistency) For $y \in \mathcal{Y}$ fixed, if $D_\phi(x_n, y) \to 0$, then $x_n \to y$,*

*then for any $\epsilon > 0$, there exists $\delta > 0$ such that for all $f_s \in \mathcal{F}$ that satisfy*

$$\mathbb{E}_X \left[D_\phi \left(f_s \left(h_s(X)\right), f_w \left(h_w(X)\right)\right)\right] \leq \inf_{f \in \mathcal{F}} \mathbb{E}_X \left[D_\phi \left(f \left(h_s(X)\right), f_w \left(h_w(X)\right)\right)\right] + \delta,$$

*we have*

$$\mathbb{E}_X \left[D_\phi \left(g(X), f_s \left(h_s(X)\right)\right)\right] \leq \mathbb{E}_X \left[D_\phi \left(g(X), f_w \left(h_w(X)\right)\right)\right] - \mathbb{E}_X \left[D_\phi \left(f_s \left(h_s(X)\right), f_w \left(h_w(X)\right)\right)\right] + \epsilon. \tag{11}$$

**Remark A.2.** *In contrast to Theorem A.2, our bounds in Theorem 4.1 are derived without relying on any of the three assumptions mentioned earlier: realizability, convexity, and sequential consistency. Moreover, the conditions under which the remainder term $\epsilon$ vanishes differ between Eq. (11) and Eq. (4). Specifically, in the case of Eq. (11), the $\epsilon$ term disappears when the student model satisfies*

$$f_s = \arg\min_{f \in \mathcal{F}} \mathbb{E}_X \left[D_\phi \left(f \left(h_s(X)\right), f_w \left(h_w(X)\right)\right)\right].$$

*By contrast, the $\epsilon_1$ term in Eq. (4) vanishes under the condition that the student model is close to its posterior mean teacher, as established in our theoretical analysis.*

# B. Omitted Proofs in Section 4 and Additional Results

## B.1. Proof of Theorem 4.1

Theorem 4.1 is restated as follows.

**Theorem 4.1.** *Let $\zeta_1 = \sqrt{\mathbb{E} \|g(X) - f_{W'}(X)\|^2}$ and $\zeta_2 = \sqrt{\mathbb{E} \|g^*(X) - f_{W'}^*(X)\|^2}$. The following inequalities hold,*

$$\mathbb{E}\left[D_\phi\left(g(X), f_{W'}(X)\right)\right] \leq \mathbb{E}\left[D_\phi\left(g(X), f_W(X)\right)\right] \\ - \mathbb{E}\left[D_\phi\left(f_{W'}(X), f_W(X)\right)\right] + \epsilon_1, \tag{4}$$

$$\mathbb{E}\left[D_\phi\left(f_{W'}(X), g(X)\right)\right] \leq \mathbb{E}\left[D_\phi\left(f_W(X), g(X)\right)\right] \\ - \mathbb{E}\left[D_\phi\left(f_W(X), f_{W'}(X)\right)\right] + \epsilon_2. \tag{5}$$

*where $\epsilon_1 = \zeta_1 \sqrt{\mathbb{E} \|f_{W'}^*(X) - \mathbb{E}[f_W^*(X) \mid W', X]\|^2}$ and $\epsilon_2 = \zeta_2 \sqrt{\mathbb{E} \|f_{W'}(X) - \mathbb{E}[f_W(X) \mid W', X]\|^2}$.*

*Proof.* We first prove Eq. (4).

For any given test instance $x$ and a fixed student model $f_{w'}$, by Eq. (3), we have the following conditional bias-variance decomposition for the weak teacher model,

$$\mathbb{E}_{W|w'}\left[D_\phi\left(g(x), f_W(x)\right)\right] = D_\phi\left(g(x), \mathcal{E}_{W|w'}[f_W(x)]\right) + \mathbb{E}_{W|w'}\left[D_\phi\left(\mathcal{E}_{W|w'}[f_W(x)], f_W(x)\right)\right]. \tag{12}$$

Similarly, we also have

$$\mathbb{E}_{W|w'}\left[D_\phi\left(f_{w'}(x), f_W(x)\right)\right] = D_\phi\left(f_{w'}(x), \mathcal{E}_{W|w'}[f_W(x)]\right) + \mathbb{E}_{W|w'}\left[D_\phi\left(\mathcal{E}_{W|w'}[f_W(x)], f_W(x)\right)\right]. \tag{13}$$

Notice that both Eq. (12) and Eq. (13) contain the conditional dual variance term of weak model, namely $\mathbb{E}_{W|w'}\left[D_\phi\left(\mathcal{E}_{W|w'}[f_W(x)], f_W(x)\right)\right]$, so combining these two equations give us

$$D_\phi\left(g(x), \mathcal{E}_{W|w'}[f_W(x)]\right) \\ = \mathbb{E}_{W|w'}\left[D_\phi\left(g(x), f_W(x)\right)\right] - \mathbb{E}_{W|w'}\left[D_\phi\left(f_{w'}(x), f_W(x)\right)\right] + D_\phi\left(f_{w'}(x), \mathcal{E}_{W|w'}[f_W(x)]\right).$$

Then, by taking expectation over $W'$ for both sides, we obtain

$$\mathbb{E}_{W'}\left[D_\phi\left(g(x), \mathcal{E}_{W|W'}[f_W(x)]\right)\right] \\ = \mathbb{E}_W\left[D_\phi\left(g(x), f_W(x)\right)\right] - \mathbb{E}_{W',W}\left[D_\phi\left(f_{W'}(x), f_W(x)\right)\right] + \mathbb{E}_{W'}\left[D_\phi\left(f_{W'}(x), \mathcal{E}_{W|W'}[f_W(x)]\right)\right]. \tag{14}$$

We now further decompose the LHS in Eq. (14) by invoking Eq. (1) here,.

$$\mathbb{E}_{W'}\left[D_\phi\left(g(x), \mathcal{E}_{W|w'}[f_W(x)]\right)\right] = \mathbb{E}_{W'}\left[D_\phi\left(g(x), f_{W'}(x)\right)\right] + \mathbb{E}_{W'}\left[D_\phi\left(f_{W'}(x), \mathcal{E}_{W|w'}[f_W(x)]\right)\right] \\ - \mathbb{E}_{W'}\left\langle\mathbb{E}_{W|W'}[f_W^*(x)] - f_{W'}^*(x), g(x) - f_{W'}(x)\right\rangle. \tag{15}$$

Plugging Eq. (15) into Eq. (14), taking expectation over $X$ for both sides and re-arranging terms, we have

$$\mathbb{E}_{X,W'}\left[D_\phi\left(g(X), f_{W'}(X)\right)\right] \\ = \mathbb{E}_{X,W}\left[D_\phi\left(g(X), f_W(X)\right)\right] - \mathbb{E}_{X,W',W}\left[D_\phi\left(f_{W'}(X), f_W(X)\right)\right] \\ + \mathbb{E}_{X,W'}\left\langle\mathbb{E}_{W|W'}[f_W^*(X)] - f_{W'}^*(X), g(X) - f_{W'}(X)\right\rangle \tag{16} \\ \leq \mathbb{E}_{X,W}\left[D_\phi\left(g(X), f_W(X)\right)\right] - \mathbb{E}_{X,W',W}\left[D_\phi\left(f_{W'}(X), f_W(X)\right)\right] \\ + \sqrt{\left(\mathbb{E}_{X,W'}\left\langle\mathbb{E}_{W|W'}[f_W^*(X)] - f_{W'}^*(X), g(X) - f_{W'}(X)\right\rangle\right)^2} \\ \leq \mathbb{E}_{X,W}\left[D_\phi\left(g(X), f_W(X)\right)\right] - \mathbb{E}_{X,W',W}\left[D_\phi\left(f_{W'}(X), f_W(X)\right)\right] \\ + \sqrt{\mathbb{E}_{X,W'}\left\|\mathbb{E}_{W|W'}[f_W^*(X)] - f_{W'}^*(X)\right\|^2}\sqrt{\mathbb{E}_{X,W'}\|g(X) - f_{W'}(X)\|^2}, \tag{17}$$

where the last inequality is by Cauchy-Schwarz inequality. Hence, the first inequality in the theorem has been proved.

The second inequality, namely Eq. (5), can be proved by following the same developments. Similar to Eq. (14), it's easy to see that we also have

$$
\begin{aligned}
& \mathbb{E}_{W'}\left[\mathrm{D}_\phi\left(\mathbb{E}_{W|W'}\left[f_W(x)\right], g(x)\right)\right] \\
= & \mathbb{E}_W\left[\mathrm{D}_\phi\left(f_W(x), g(x)\right)\right] - \mathbb{E}_{W',W}\left[\mathrm{D}_\phi\left(f_W(x), f_{W'}(x)\right)\right] + \mathbb{E}_{W'}\left[\mathrm{D}_\phi\left(\mathbb{E}_{W|W'}\left[f_W(x)\right], f_{W'}(x)\right)\right].
\end{aligned}
$$

We then decompose the LHS above by using Eq. (1),

$$
\begin{aligned}
\mathbb{E}_{W'}\left[\mathrm{D}_\phi\left(\mathbb{E}_{W|W'}\left[f_W(x)\right], g(x)\right)\right] = & \mathbb{E}_{W'}\left[\mathrm{D}_\phi\left(\mathbb{E}_{W|W'}\left[f_W(x)\right], f_{W'}(x)\right)\right] + \mathbb{E}_{W'}\left[\mathrm{D}_\phi\left(f_{W'}(x), g(x)\right)\right] \\
& - \mathbb{E}_{W'}\left\langle g^*(x) - f_{W'}^*(x), \mathbb{E}_{W|W'}\left[f_W(x)\right] - f_{W'}(x)\right\rangle.
\end{aligned}
$$

Consequently, we have

$$
\begin{aligned}
& \mathbb{E}_{X,W'}\left[\mathrm{D}_\phi\left(f_{W'}(X), g(X)\right)\right] \\
= & \mathbb{E}_{X,W}\left[\mathrm{D}_\phi\left(f_W(X), g(X)\right)\right] - \mathbb{E}_{X,W',W}\left[\mathrm{D}_\phi\left(f_W(X), f_{W'}(X)\right)\right] \\
& + \mathbb{E}_{X,W'}\left\langle g^*(X) - f_{W'}^*(X), \mathbb{E}_{W|W'}\left[f_W(X)\right] - f_{W'}(X)\right\rangle \\
\leq & \mathbb{E}_{X,W}\left[\mathrm{D}_\phi\left(f_W(X), g(X)\right)\right] - \mathbb{E}_{X,W',W}\left[\mathrm{D}_\phi\left(f_W(X), f_{W'}(X)\right)\right] \\
& + \sqrt{\mathbb{E}_{X,W'}\left\|g^*(X) - f_{W'}^*(X)\right\|^2}\sqrt{\mathbb{E}_{X,W'}\left\|f_{W'}(X) - \mathbb{E}_{W|W'}\left[f_W(X)\right]\right\|^2}.
\end{aligned}
\tag{18}
$$

This completes the proof. $\qquad\square$

## B.2. Additional Result: Expected Misfit for Arbitrary Two Models

**Theorem B.1.** *For arbitrary two models $f_w$ and $f_{w'}$, the following inequalities hold for any Bregman divergence,*

$$
\mathbb{E}_{W'}\left[\mathrm{D}_\phi\left(g(X), f_{W'}(X)\right)\right] \leq \mathbb{E}_W\left[\mathrm{D}_\phi\left(g(X), f_W(X)\right)\right] - \mathbb{E}_{P_X P_{W'} P_W}\left[\mathrm{D}_\phi\left(f_{W'}(X), f_W(X)\right)\right] + \epsilon_1,
\tag{19}
$$

$$
\mathbb{E}_{W'}\left[\mathrm{D}_\phi\left(f_{W'}(X), g(X)\right)\right] \leq \mathbb{E}_W\left[\mathrm{D}_\phi\left(f_W(X), g(X)\right)\right] - \mathbb{E}_{P_X P_{W'} P_W}\left[\mathrm{D}_\phi\left(f_W(X), f_{W'}(X)\right)\right] + \epsilon_2,
\tag{20}
$$

*where $\epsilon_1 = \sqrt{\mathbb{E}_{W'}\left\|f_{W'}^*(X) - \mathbb{E}_W\left[f_W^*(X)\right]\right\|^2}\sqrt{\mathbb{E}\left\|g(X) - f_{W'}(X)\right\|^2}$ and $\epsilon_2 = \sqrt{\mathbb{E}_{W'}\left\|f_{W'}(X) - \mathbb{E}_W\left[f_W(X)\right]\right\|^2}\sqrt{\mathbb{E}_{W'}\left\|g^*(X) - f_{W'}^*(X)\right\|^2}$.*

**Remark B.1.** *Note that, unlike Theorem 4.1, the misfit terms in Theorem B.1 are defined under the expectation with respect to the product of the marginal distributions of the teacher and student models, i.e., $(W, W') \sim P_W P_{W'}$. In contrast, Theorem 4.1 considers the joint distribution $(W, W') \sim P_{W,W'}$, thereby capturing potential dependencies between the models. As a result, Theorem B.1 completely ignores any such dependencies, making it applicable even when the teacher and student models are independently drawn. Additionally, the conditions under which the remainder terms $\epsilon_1$ and $\epsilon_2$ vanish are now characterized by the student model being sufficiently close to the expected teacher model (or the dual expected teacher). Finally, observe that in Theorem B.1, the misfit terms remain nonzero even when $P_{W'}$ is the same $P_W$ as $W'$ is treated as an independent copy of $W$ in this case.*

*Proof of Theorem B.1.* We first prove the first inequality, i.e. Eq. (19).

For any given test instance $x$ and a fixed student model $f_{w'}$, by Eq. (3), we have the following bias-variance decomposition for the weak teacher model,

$$
\mathbb{E}_W\left[\mathrm{D}_\phi\left(g(x), f_W(x)\right)\right] = \mathrm{D}_\phi\left(g(x), \mathcal{E}_W\left[f_W(x)\right]\right) + \mathbb{E}_W\left[\mathrm{D}_\phi\left(\mathcal{E}_W\left[f_W(x)\right], f_W(x)\right)\right].
\tag{21}
$$

Similarly, for the same $x$ and the student model $f_{w'}$, we also have

$$
\mathbb{E}_W\left[\mathrm{D}_\phi\left(f_{w'}(x), f_W(x)\right)\right] = \mathrm{D}_\phi\left(f_{w'}(x), \mathcal{E}_W\left[f_W(x)\right]\right) + \mathbb{E}_W\left[\mathrm{D}_\phi\left(\mathcal{E}_W\left[f_W(x)\right], f_W(x)\right)\right].
\tag{22}
$$

Notice that both Eq. (21) and Eq. (22) contain the dual variance term of weak model, namely $\mathbb{E}_W \left[ \mathrm{D}_\phi \left( \mathcal{E}_W \left[ f_W(x) \right], f_W(x) \right) \right]$, so combining these two equations give us

$$\mathrm{D}_\phi \left( g(x), \mathcal{E}_W \left[ f_W(x) \right] \right) = \mathbb{E}_W \left[ \mathrm{D}_\phi \left( g(x), f_W(x) \right) \right] - \mathbb{E}_W \left[ \mathrm{D}_\phi \left( f_{w'}(x), f_W(x) \right) \right] \\ + \mathrm{D}_\phi \left( f_{w'}(x), \mathcal{E}_W \left[ f_W(x) \right] \right). \tag{23}$$

We now further decompose the LHS above by invoking Eq. (1) here.

$$\mathrm{D}_\phi \left( g(x), \mathcal{E}_W \left[ f_W(x) \right] \right) = \mathrm{D}_\phi \left( g(x), f_{w'}(x) \right) + \mathrm{D}_\phi \left( f_{w'}(x), \mathcal{E}_W \left[ f_W(x) \right] \right) \\ - \left\langle \mathbb{E}_W \left[ f_W^*(x) \right] - f_{w'}^*(x), g(x) - f_{w'}(x) \right\rangle. \tag{24}$$

Then, by plugging Eq. (24) into Eq. (23), canceling the term $\mathrm{D}_\phi \left( f_{w'}(x), \mathcal{E}_W \left[ f_W(x) \right] \right)$ and taking expectation over $W', X$ for both sides, we obtain

$$\mathbb{E}_{X,W'} \left[ \mathrm{D}_\phi \left( g(X), f_{W'}(X) \right) \right] \\ = \mathbb{E}_{X,W} \left[ \mathrm{D}_\phi \left( g(X), f_W(X) \right) \right] - \mathbb{E}_{X,W',W} \left[ \mathrm{D}_\phi \left( f_{W'}(X), f_W(X) \right) \right] \\ + \mathbb{E}_{X,W'} \left\langle \mathbb{E}_W \left[ f_W^*(X) \right] - f_{W'}^*(X), g(X) - f_{W'}(X) \right\rangle \\ \leq \mathbb{E}_{X,W} \left[ \mathrm{D}_\phi \left( g(X), f_W(X) \right) \right] - \mathbb{E}_{X,W',W} \left[ \mathrm{D}_\phi \left( f_{W'}(X), f_W(X) \right) \right] \\ + \sqrt{ \left( \mathbb{E}_{X,W'} \left\langle \mathbb{E}_W \left[ f_W^*(X) \right] - f_{W'}^*(X), g(X) - f_{W'}(X) \right\rangle \right)^2 } \\ \leq \mathbb{E}_{X,W} \left[ \mathrm{D}_\phi \left( g(X), f_W(X) \right) \right] - \mathbb{E}_{X,W',W} \left[ \mathrm{D}_\phi \left( f_{W'}(X), f_W(X) \right) \right] \\ + \sqrt{ \mathbb{E}_{X,W'} \left\| \mathbb{E}_W \left[ f_W^*(X) \right] - f_{W'}^*(X) \right\|^2 } \sqrt{ \mathbb{E}_{X,W'} \left\| g(X) - f_{W'}(X) \right\|^2 }, \tag{25}$$

where the last inequality is by Cauchy-Schwarz inequality. Hence, the first inequality in the theorem has been proved.

The second inequality, namely Eq. (20), can be proved by following the same developments. Similar to Eq. (23), it's easy to see that we also have

$$\mathrm{D}_\phi \left( \mathbb{E}_W \left[ f_W(x) \right], g(x) \right) \\ = \mathbb{E}_W \left[ \mathrm{D}_\phi \left( f_W(x), g(x) \right) \right] - \mathbb{E}_W \left[ \mathrm{D}_\phi \left( f_W(x), f_{w'}(x) \right) \right] + \mathrm{D}_\phi \left( \mathbb{E}_W \left[ f_W(x) \right], f_{w'}(x) \right).$$

We then decompose the LHS above by using Eq. (1),

$$\mathrm{D}_\phi \left( \mathbb{E}_W \left[ f_W(x) \right], g(x) \right) = \mathrm{D}_\phi \left( \mathbb{E}_W \left[ f_W(x) \right], f_{w'}(x) \right) + \mathrm{D}_\phi \left( f_{w'}(x), g(x) \right) \\ - \left\langle g^*(x) - f_{w'}^*(x), \mathbb{E}_W \left[ f_W(x) \right] - f_{w'}(x) \right\rangle.$$

Consequently, we have

$$\mathbb{E}_{X,W'} \left[ \mathrm{D}_\phi \left( f_{W'}(X), g(X) \right) \right] \\ = \mathbb{E}_{X,W} \left[ \mathrm{D}_\phi \left( f_W(X), g(X) \right) \right] - \mathbb{E}_{X,W',W} \left[ \mathrm{D}_\phi \left( f_W(X), f_{W'}(X) \right) \right] \\ + \mathbb{E}_{X,W'} \left\langle g^*(X) - f_{W'}^*(X), \mathbb{E}_W \left[ f_W(X) \right] - f_{W'}(X) \right\rangle \\ \leq \mathbb{E}_{X,W} \left[ \mathrm{D}_\phi \left( f_W(X), g(X) \right) \right] - \mathbb{E}_{X,W',W} \left[ \mathrm{D}_\phi \left( f_W(X), f_{W'}(X) \right) \right] \\ + \sqrt{ \mathbb{E}_{X,W'} \left\| g^*(X) - f_{W'}^*(X) \right\|^2 } \sqrt{ \mathbb{E}_{X,W'} \left\| f_{W'}(X) - \mathbb{E}_W \left[ f_W(X) \right] \right\|^2 }. \tag{26}$$

This completes the proof. □

### B.3. Proof of Corollary 4.1

We first restate Corollary 4.1 as follows.

**Corollary 4.1.** *Under the teacher-student setting, if $f_{W'}(x) = \mathcal{E}\left[f_W(x)|W'\right]$ for $\forall x \in \mathcal{X}$, then*

$$
\begin{aligned}
\mathbb{E}\left[\mathrm{D}_\phi\left(g(X), f_{W'}(X)\right)\right] = {} & \mathbb{E}\left[\mathrm{D}_\phi\left(g(X), f_W(X)\right)\right] \\
& - \mathbb{E}\left[\mathrm{D}_\phi\left(\mathcal{E}\left[f_W(X)|W', X\right], f_W(X)\right)\right].
\end{aligned}
\tag{6}
$$

*Furthermore, if $f_{W'}(x) = \mathbb{E}\left[f_W(x)|W'\right]$ for $\forall x \in \mathcal{X}$, then*

$$
\begin{aligned}
\mathbb{E}\left[\mathrm{D}_\phi\left(f_{W'}(X), g(X)\right)\right] = {} & \mathbb{E}\left[\mathrm{D}_\phi\left(f_W(X), g(X)\right)\right] \\
& - \mathbb{E}\left[\mathrm{D}_\phi\left(f_W(X), \mathbb{E}\left[f_W(X)|W', X\right]\right)\right].
\end{aligned}
\tag{7}
$$

*Proof.* If $f_{W'}(x) = \mathcal{E}\left[f_W(x)|W'\right]$ for $\forall x \in \mathcal{X}$, then by definition, $f^*_{W'}(x) = \mathbb{E}\left[f_W(x)|W'\right]$, which directly implies $\epsilon_1 = 0$. Furthermore, since $f^*_{W'}(x) = \mathbb{E}\left[f_W(x)|W'\right]$, the last term in Eq. (15) becomes zero so there is no need to apply the Cauchy-Schwarz inequality in the proof of Theorem 4.1 to obtain $\epsilon_1$. Consequently, the following equality holds directly from Theorem 4.1:

$$
\mathbb{E}_{X,W'}\left[\mathrm{D}_\phi\left(g(X), f_{W'}(X)\right)\right] = \mathbb{E}_{X,W}\left[\mathrm{D}_\phi\left(g(X), f_W(X)\right)\right] - \mathbb{E}_{X,W',W}\left[\mathrm{D}_\phi\left(\mathcal{E}_W\left[f_W(X)|W', X\right], f_W(X)\right)\right].
$$

Similarly, if $f_{W'}(x) = \mathbb{E}\left[f_W(x)|W'\right]$ for $\forall x \in \mathcal{X}$, we obtain the following by analogous reasoning:

$$
\mathbb{E}_{X,W'}\left[\mathrm{D}_\phi\left(f_{W'}(X), g(X)\right)\right] = \mathbb{E}_{X,W}\left[\mathrm{D}_\phi\left(f_W(X), g(X)\right)\right] - \mathbb{E}_{X,W',W}\left[\mathrm{D}_\phi\left(f_W(X), \mathbb{E}_W\left[f_W(X)|W', X\right]\right)\right].
\tag{27}
$$

This completes the proof. $\qquad\square$

## C. Omitted Proofs in Section 5

### C.1. Proof of Corollary 5.1

We first restate Corollary 5.1.

**Corollary 5.1.** *Let $\phi(x) = \|x\|^2$, then the following inequality holds,*

$$
\begin{aligned}
\mathbb{E}\|g(X) - f_{W'}(X)\|^2 \leq {} & \mathbb{E}\|g(X) - f_W(X)\|^2 \\
& - \mathbb{E}\|f_{W'}(X) - f_W(X)\|^2 + \epsilon_2,
\end{aligned}
\tag{8}
$$

*where $\epsilon_2 = 2\zeta_1\sqrt{\mathbb{E}\left\|f_{W'}(X) - \mathbb{E}\left[f_W(X) \mid W', X\right]\right\|^2}$.*

*Proof.* Let $\phi(x) = \|x\|^2$ so $x^* = \nabla\phi(x) = 2x$ and $\mathrm{D}_\phi(x, y) = \|x - y\|^2$. Clearly, we have $\epsilon_1 = \epsilon_2$ and $\mathrm{D}_\phi(x, y) = \mathrm{D}_\phi(y, x)$ in this case. Then, by Theorem 4.1, it is easy to see that

$$
\begin{aligned}
\mathbb{E}\|g(X) - f_{W'}(X)\|^2 \leq {} & \mathbb{E}\|g(X) - f_W(X)\|^2 - \mathbb{E}\|f_{W'}(X) - f_W(X)\|^2 \\
& + 2\sqrt{\mathbb{E}\left\|f_{W'}(X) - \mathbb{E}\left[f_W(X) \mid W', X\right]\right\|^2}\sqrt{\mathbb{E}\|g(X) - f_{W'}(X)\|^2}.
\end{aligned}
$$

This completes the proof. $\qquad\square$

### C.2. Proof of Theorem 5.1

We first present the following lemma.

**Lemma C.1.** *Let $C = \mathbf{X}'\mathbf{X}'^\top$, where $\mathbf{X}' \in \mathbb{R}^{d_w \times n}$ has i.i.d. entries $\mathcal{N}(0, 1/n)$, and assume $n/d_w \to \gamma_2 \in (0, 1)$. Let $E_{d_w}$ be a sequence of symmetric matrices independent of $C$ such that $\|E_{d_w}\|_{\mathrm{op}} \to 0$. Then, for every fixed $\eta > 0$ and $\widetilde{\eta} = \eta/\gamma_1 \to 0$,*

$$
\widetilde{\eta}^2 \frac{1}{d_w}\mathrm{tr}\left[(\widetilde{\eta}\mathbf{I} + (\mathbf{I} + E_{d_w})C)^{-1}(\widetilde{\eta}\mathbf{I} + C(\mathbf{I} + E_{d_w}))^{-1}\right] - \widetilde{\eta}^2\frac{1}{d_w}\mathrm{tr}\left[(\widetilde{\eta}\mathbf{I} + C)^{-2}\right] \xrightarrow{\mathbb{P}} 0.
$$

*Proof of Lemma C.1.* Because $\|E_{d_w}\|_{\mathrm{op}} \to 0$ and $\|C\|_{\mathrm{op}} = O_{\mathbb{P}}(1)$, we have

$$\|(\mathbf{I} + E_{d_w})C - C\|_{\mathrm{op}} = o_{\mathbb{P}}(1).$$

Hence, for each fixed $\widetilde{\eta} > 0$,

$$\left\| (\widetilde{\eta}\mathbf{I} + (\mathbf{I} + E_{d_w})C)^{-1} - (\widetilde{\eta}\mathbf{I} + C)^{-1} \right\|_{\mathrm{op}} \to 0.$$

Although $\widetilde{\eta} = \eta/\gamma_1 \to 0$, the final quantity is multiplied by $\widetilde{\eta}^2$. The zero-eigenvalue subspace of $C$ is unaffected by the perturbation $E_{d_w} = o_{\mathrm{op}}(1)$, and the nonzero spectral part is bounded away from zero in the Marchenko–Pastur limit when $\gamma_2 < 1$. Therefore,

$$\eta^2 \frac{1}{d_w} \operatorname{tr}\left[ M_Q^{-1}(M_Q^{\top})^{-1} \right] - \widetilde{\eta}^2 \frac{1}{d_w} \operatorname{tr}\left[ (\widetilde{\eta}I + C)^{-2} \right] \to 0.$$

This completes the proof. $\qquad\square$

We now restate Theorem 5.1.

**Theorem 5.1.** *Assume $W \sim \mathcal{N}(\mathbf{m}, B\mathbf{I}_{d_w})$ where $\mathbf{m} \in \mathbb{R}^{d_w}$ is independent of $W_1'$ and $\mathbf{X}'$, and $\|\mathbf{m}\|^2 \leq c$ for some fixed constant c. Suppose that $\frac{d_s}{d_w} = \gamma_1 \to \infty$ and $\frac{n}{d_w} \to \gamma_2 \in (0,1)$ as $n, d_w, d_s \to \infty$. Then, for fixed $\eta > 0$, as $n, d_s, d_w \to \infty$, the trained ridge student satisfies*

$$\mathbb{E}\|f_{W'}(X) - f_W(X)\|^2 \to B(1 - \gamma_2).$$

*Proof.* First, the solution to the ridge regression of student model is

$$W_2'^{\mathrm{opt}} = \left( W_1'\mathbf{X}'\mathbf{X}'^T W_1'^T + \eta\mathbf{I} \right)^{-1} W_1'\mathbf{X}'\mathbf{X}'^T W.$$

Hence the trained student model becomes

$$f_{W'}(x) = x^\top W_1'^\top W_2'^{\mathrm{opt}} = x^\top \Theta W,$$

where $\Theta = W_1'^\top \left( W_1'\mathbf{X}'\mathbf{X}'^\top W_1'^\top + \eta\mathbf{I} \right)^{-1} W_1'\mathbf{X}'\mathbf{X}'^\top \in \mathbb{R}^{d_w \times d_w}$.

Define

$$Z = W_1'\mathbf{X}' \in \mathbb{R}^{d_s \times n}, \qquad Q = W_1'^\top W_1' \in \mathbb{R}^{d_w \times d_w}, \qquad K = Z^\top Z = \mathbf{X}'^\top Q \mathbf{X}' \in \mathbb{R}^{n \times n}.$$

By the matrix identity $(AA^\top + \eta\mathbf{I})^{-1}A = A(A^\top A + \eta\mathbf{I})^{-1}$, we have

$$W_1'^\top (ZZ^\top + \eta\mathbf{I})^{-1}Z = Q\mathbf{X}'(K + \eta\mathbf{I})^{-1}. \tag{28}$$

Furthermore, by the Woodbury matrix identity $(\mathbf{I} + UV)^{-1} = \mathbf{I} - U(I + VU)^{-1}V$, we can see that $\mathbf{I} - Q\mathbf{X}'(K + \eta\mathbf{I})^{-1}\mathbf{X}'^\top = \eta(\eta\mathbf{I} + Q\mathbf{X}'\mathbf{X}'^\top)^{-1}$, namely

$$Q\mathbf{X}'(K + \eta\mathbf{I})^{-1}\mathbf{X}'^\top = \mathbf{I} - \eta(\eta\mathbf{I} + Q\mathbf{X}'\mathbf{X}'^\top)^{-1}. \tag{29}$$

Combining Eq. (28) and Eq. (29) together, we can obtain

$$\Theta = W_1'^\top \left( ZZ^\top + \eta\mathbf{I} \right)^{-1} Z\mathbf{X}'^\top = \mathbf{I} - \eta(\eta\mathbf{I} + Q\mathbf{X}'\mathbf{X}'^\top)^{-1}. \tag{30}$$

Now consider the expected misfit. Notice that decomposition,

$$
\begin{aligned}
&\mathbb{E}_{X,W',W} \left\| f_{W'}(X) - f_W(X) \right\|^2 \\
=& \mathbb{E}_{X,\Theta,W} \left\| f_{W'}(X) - \mathbb{E}_{W'|W}[f_{W'}(X)] \right\|^2 + \mathbb{E}_{X,\Theta,W} \left\| \mathbb{E}_{W'|W}[f_{W'}(X)] - f_W(X) \right\|^2 && (31) \\
=& \mathbb{E}_{X,\Theta,W} \left\| X^T \Theta W - \mathbb{E}_\Theta[X^T \Theta W] \right\|^2 + \mathbb{E}_{X,W} \left\| \mathbb{E}_\Theta[X^\top \Theta W] - X^\top W \right\|^2 \\
=& \mathbb{E}_{X,W,\Theta} \left\| X^\top (\Theta - \mathbb{E}_\Theta[\Theta]) W \right\|^2 + \mathbb{E}_{X,W} \left\| X^\top (\mathbb{E}_\Theta[\Theta] - \mathbf{I}) W \right\|^2 \\
=& \frac{1}{d_w} \mathbb{E}_{W,\Theta} \left\| (\Theta - \mathbb{E}_\Theta[\Theta]) W \right\|^2 + \frac{1}{d_w} \mathbb{E}_W \left\| (\mathbb{E}_\Theta[\Theta] - \mathbf{I}) W \right\|^2 && (32) \\
=& \frac{B + \|\mathbf{m}\|^2/d_w}{d_w} \left( \mathbb{E}_\Theta \|\Theta - \mathbb{E}_\Theta[\Theta]\|^2 + \|\mathbb{E}_\Theta[\Theta] - \mathbf{I}\|^2 \right) && (33) \\
=& \frac{B + \|\mathbf{m}\|^2/d_w}{d_w} \mathbb{E}_\Theta \|\Theta - \mathbf{I}\|^2, && (34) \\
=& \left( B + \frac{\|\mathbf{m}\|^2}{d_w} \right) \eta^2 \mathbb{E}\left[ \frac{1}{d_w} \operatorname{tr} \left( (\eta\mathbf{I} + Q\mathbf{X}'\mathbf{X}'^\top)^{-1} (\eta\mathbf{I} + \mathbf{X}'\mathbf{X}'^\top Q)^{-1} \right) \right], && (35)
\end{aligned}
$$

where Eq. (31) is by the bias-variance decomposition of $\mathbb{E}_{X,W',W} \|f_{W'}(X) - f_W(X)\|^2$, Eq. (32) is due to the fact that $\mathbb{E}[XX^T] = \frac{1}{d_w}\mathbf{I}$ and Eq. (33) is by $\mathbb{E}[WW^\top] = \operatorname{Cov}(W) + \mathbb{E}[W]\mathbb{E}[W]^\top = B\mathbf{I} + \mathbf{mm}^\top$ and the rotational invariance property of Gaussian (Ba et al., 2020, Lemma 10).

We now use the additional assumption that the student random features are asymptotically isotropic. Since $Q = W_1'^\top W_1'$, and $W_1'$ has entries $\mathcal{N}(0, 1/d_w)$, we can write $W_1' = \frac{1}{\sqrt{d_w}}G$, where $G \in \mathbb{R}^{d_s \times d_w}$ is from $\mathcal{N}(0, \mathbf{I})$. Hence

$$
Q = \frac{1}{d_w} G^\top G = \gamma_1 \left( \frac{1}{d_s} G^\top G \right).
$$

The empirical law of $Q$ converges a.s. to the Marchenko–Pastur law $H_{\gamma_1}$ supported on $\left( (\sqrt{\gamma_1} - 1)^2, (\sqrt{\gamma_1} + 1)^2 \right)$ with $\gamma_1 > 1$ when $d_w \to \infty$ (Marčenko & Pastur, 1967) (note that $Q$ is the spectrum of the standard MP matrix $(1/d_s)G^\top G$ scaled by $\gamma_1$). In addition, because $d_w/d_s = 1/\gamma_1 \to 0$, standard operator-norm concentration for Gaussian sample covariance matrices gives

$$
\left\| \frac{1}{d_s} G^\top G - \mathbf{I} \right\|_{\mathrm{op}} \xrightarrow{a.s.} 0.
$$

Therefore,

$$
Q = \gamma_1 (\mathbf{I} + E_{d_w}), \qquad \text{with } \|E_{d_w}\|_{\mathrm{op}} \to 0.
$$

Let $C = \mathbf{X}'\mathbf{X}'^\top$, and let $\widetilde{\eta} = \frac{\eta}{\gamma_1}$. Then define

$$
M_Q \triangleq \eta\mathbf{I} + QC = \gamma_1 \left( \widetilde{\eta}\mathbf{I} + (\mathbf{I} + E_{d_w})C \right), \qquad \text{and } M_0 \triangleq \eta\mathbf{I} + \gamma_1 C = \gamma_1 (\widetilde{\eta}\mathbf{I} + C).
$$

Therefore,

$$
M_Q^{-1} = \frac{1}{\gamma_1} \left( \widetilde{\eta}\mathbf{I} + (\mathbf{I} + E_{d_w})C \right)^{-1}, \qquad \text{and } M_0^{-1} = \frac{1}{\gamma_1} (\widetilde{\eta}\mathbf{I} + C)^{-1}.
$$

Multiplying by the prefactor $\eta^2$ in the misfit (i.e. in Eq. (35)) gives

$$
\eta^2 M_Q^{-1}(M_Q^\top)^{-1} = \widetilde{\eta}^2 \left( \widetilde{\eta}\mathbf{I} + (\mathbf{I} + E_{d_w})C \right)^{-1} \left( \widetilde{\eta}\mathbf{I} + C(\mathbf{I} + E_{d_w}) \right)^{-1}.
$$

Using Lemma C.1, we know that, when $\gamma_1 \to \infty$,

$$
\eta^2 \frac{1}{d_w} \operatorname{tr} \left( (\eta\mathbf{I} + Q\mathbf{X}'\mathbf{X}'^\top)^{-1} (\eta\mathbf{I} + \mathbf{X}'\mathbf{X}'^\top Q)^{-1} \right) - \widetilde{\eta}^2 \frac{1}{d_w} \operatorname{tr} \left[ (\widetilde{\eta}I + C)^{-2} \right] \to 0. \tag{36}
$$

It remains to evaluate $\widetilde{\eta}^2 \frac{1}{d_w} \operatorname{tr} \left[ (\widetilde{\eta}\mathbf{I} + C)^{-2} \right]$ as $d_w, n \to \infty$ and $\widetilde{\eta} \to 0$.

Since $C = \mathbf{X}'\mathbf{X}'^\top$ has rank at most $n$, and $\frac{n}{d_w} \to \gamma_2 < 1$, the nullspace of $C$ has dimension

$$d_w - n = d_w(1 - \gamma_2) + o(d_w).$$

Let $\{\lambda_i\}_{i=1}^{d_w}$ be the eigenvalues of $C$. Then

$$\widetilde{\eta}^2 \frac{1}{d_w} \operatorname{tr}\left[(\widetilde{\eta}\mathbf{I} + C)^{-2}\right] = \frac{1}{d_w} \sum_{i=1}^{d_w} \frac{\widetilde{\eta}^2}{(\widetilde{\eta} + \lambda_i)^2}.$$

The zero eigenvalues contribute exactly

$$\frac{d_w - n}{d_w} \to 1 - \gamma_2.$$

The nonzero eigenvalues contribute

$$\frac{1}{d_w} \sum_{\lambda_i > 0} \frac{\widetilde{\eta}^2}{(\widetilde{\eta} + \lambda_i)^2}.$$

By the Marchenko–Pastur law, the empirical distribution of the nonzero eigenvalues of $C$ has support bounded away from zero when $\frac{n}{d_w} \to \gamma_2 \in (0, 1)$. Hence this nonzero-eigenvalue contribution vanishes as $\widetilde{\eta} \to 0$. Therefore,

$$\widetilde{\eta}^2 \frac{1}{d_w} \operatorname{tr}\left[(\widetilde{\eta}\mathbf{I} + C)^{-2}\right] \to 1 - \gamma_2. \tag{37}$$

Combining Eq. (35), Eq. (36) and Eq. (37), we obtain

$$\mathbb{E}_{X,W',W} \|f_{W'}(X) - f_W(X)\|^2 = \left(B + \frac{\|\mathbf{m}\|^2}{d_w}\right)\left[\eta^2 \mathbb{E} \frac{1}{d_w} \operatorname{tr}\left\{(\eta\mathbf{I} + QC)^{-1}(\eta\mathbf{I} + CQ)^{-1}\right\}\right] \to B(1 - \gamma_2),$$

where we used the fact that $\|\mathbf{m}\|^2 / d_w \to 0$.

This completes the proof. $\qquad\square$

### C.3. Proof of Corollary 5.2

Corollary 5.2 is restated as follows.

**Corollary 5.2.** *Under the same conditions in Theorem 5.1, we have $\epsilon_2 \to 0$.*

*Proof.* First, under the setting of Theorem 5.1, it is easy to see that the conditional variance term in Eq. (9) takes the form

$$\mathbb{E} \|f_W(x) - \mathbb{E}\left[f_W(x) \,|\, W'\right]\|^2 = \frac{\operatorname{tr}(\mathbb{E}_{W'}\operatorname{Cov}(W \mid W_1', W_2'))}{d_w}.$$

Recall the ridge head $W_2' = (ZZ^\top + \eta\mathbf{I})^{-1}Z\mathbf{X}'^\top W$, and denote $M = (ZZ^\top + \eta\mathbf{I})^{-1}Z\mathbf{X}'^\top$. Conditioning on $W_2'$ and $W_1'$ imposes the noiseless linear constraint $MW = W_2'$ on $W$.

For a Gaussian prior with a noiseless linear observation, the posterior distribution is Gaussian with

$$\operatorname{Cov}(W \mid W_1', W_2') = B\left(\mathbf{I} - \Pi_{\operatorname{col}(M^\top)}\right)$$

when $n, d_w, d_s \to \infty$, i.e., $B$ times the projector onto the orthogonal complement of $\operatorname{col}(M^\top)$.

Hence,

$$\operatorname{tr}(\mathbb{E}_{W'}\operatorname{Cov}(W \mid W_1', W_2')) = B\left(d_w - \operatorname{rank}(M)\right).$$

Consequently,

$$\frac{\operatorname{tr}(\mathbb{E}_{W'}\operatorname{Cov}(W \mid W_1', W_2'))}{d_w} = \frac{B}{d_w}(d_w - \min\{n, d_s, d_w\}) = B(1 - \min\{\gamma_2, \gamma_1, 1\}) = B(1 - \gamma_2).$$

Then, by Theorem 5.1, we know that

$$\mathbb{E}_{X,W',W} \|f_{W'}(X) - f_W(X)\|^2 \to B(1 - \gamma_2),$$

which matches the conditional variance floor $\mathbb{E} \|f_W(x) - \mathbb{E}[f_W(x)|W']\|^2$.

Plugging these equations back into Eq. (9), we have $\mathbb{E} \|f_{W'}(x) - \mathbb{E}[f_W(x)|W']\|^2 \to 0$, so $\epsilon_2 \to 0$, which completes the proof. $\qquad\square$

# D. Omitted Proofs in Section 6

## D.1. Proof of Corollary 6.1

We first restate Corollary 6.1 as follows.

**Corollary 6.1.** *Let $\phi(x) = \sum x_i \log x_i$ and define the reverse cross-entropy (RCE) as $\mathrm{RCE}(y, \hat{y}) \triangleq -\sum_{i=1}^{K} \hat{y}_i \log y_i$ for any $y, \hat{y} \in \mathcal{Y}$, then*

$$\mathbb{E}[\mathrm{CE}(g(X), f_{W'}(X))] \le \mathbb{E}[\mathrm{CE}(g(X), f_W(X))]$$
$$- \mathbb{E}[\mathrm{RCE}(f_W(X), f_{W'}(X))] + \mathbb{E}[H(f_{W'}(X))] + \epsilon_1,$$

$$\mathbb{E}[\mathrm{RCE}(g(X), f_{W'}(X))] \le \mathbb{E}[\mathrm{RCE}(g(X), f_W(X))]$$
$$- \mathbb{E}[\mathrm{CE}(f_W(X), f_{W'}(X))] + \mathbb{E}[H(f_{W'}(X))] + \epsilon_2.$$

*Proof.* Let $\phi(x) = \sum x_i \log x_i$, then $\mathrm{D}_\phi$ becomes $\mathrm{D}_{\mathrm{KL}}$. Plugging $\mathrm{D}_{\mathrm{KL}}(y\|\hat{y}) = \mathrm{CE}(y, \hat{y}) - H(y)$ into Eq. (4), we have

$$\mathbb{E}_{X,W'}[\mathrm{CE}(g(X), f_{W'}(X))] - \mathbb{E}_X[H(g(X))]$$
$$\le \mathbb{E}_{X,W}[\mathrm{CE}(g(X), f_W(X))] - \mathbb{E}_X[H(g(X))]$$
$$- \mathbb{E}_{X,W',W}[\mathrm{RCE}(f_W(X), f_{W'}(X))] + \mathbb{E}_{X,W'}[H(f_{W'}(X))] + \epsilon_1.$$

Hence,

$$\mathbb{E}_{X,W'}[\mathrm{CE}(g(X), f_{W'}(X))] \le \mathbb{E}_{X,W}[\mathrm{CE}(g(X), f_W(X))]$$
$$- \mathbb{E}_{X,W',W}[\mathrm{RCE}(f_W(X), f_{W'}(X))] + \mathbb{E}_{X,W'}[H(f_{W'}(X))] + \epsilon_1.$$

Similarly, substituting $\mathrm{D}_{\mathrm{KL}}(y\|\hat{y}) = \mathrm{CE}(y, \hat{y}) - H(y)$ into Eq. (5), we have

$$\mathbb{E}_{X,W'}[\mathrm{RCE}(g(X), f_{W'}(X))] - \mathbb{E}_{X,W'}[H(f_{W'}(X))]$$
$$\le \mathbb{E}_{X,W}[\mathrm{RCE}(g(X), f_W(X))] - \mathbb{E}_{X,W}[H(f_W(X))]$$
$$- \mathbb{E}_{X,W',W}[\mathrm{CE}(f_W(X), f_{W'}(X))] + \mathbb{E}_{X,W}[H(f_W(X))] + \epsilon_2.$$

Hence,

$$\mathbb{E}_{X,W'}[\mathrm{RCE}(g(X), f_{W'}(X))] \le \mathbb{E}_{X,W}[\mathrm{RCE}(g(X), f_W(X))]$$
$$- \mathbb{E}_{X,W',W}[\mathrm{CE}(f_W(X), f_{W'}(X))] + \mathbb{E}_{X,W'}[H(f_{W'}(X))] + \epsilon_2.$$

This completes the proof. $\qquad\square$

## D.2. Proof of Proposition 1

**Proposition 1** (Restatement). *Under the ideal conditions of Corollary 4.1 where $\epsilon_1$ and $\epsilon_2$ vanish, the performance gain in the first inequality of Corollary 6.1 coincides with Eq. (6), while the gain in the second inequality is strictly smaller than Eq. (7).*

*Proof.* If $f_{W'}(x) = \mathcal{E}\left[f_W(x)|W'\right]$ for $\forall x \in \mathcal{X}$, by Corollary 4.1 and Corollary 6.1, we have

$$
\begin{aligned}
\mathbb{E}_{X,W'}\left[\text{CE}\left(g(X), f_{W'}(X)\right)\right] =& \mathbb{E}_{X,W}\left[\text{CE}\left(g(X), f_W(X)\right)\right] - \mathbb{E}_{X,W',W}\left[\text{RCE}\left(f_W(X), \mathcal{E}\left[f_W(X)|W', X\right]\right)\right] \\
& + \mathbb{E}_{X,W'}\left[H\left(\mathcal{E}\left[f_W(X)|W', X\right]\right)\right] \\
=& \mathbb{E}_{X,W}\left[\text{CE}\left(g(X), f_W(X)\right)\right] - \mathbb{E}_{X,W,W'}\left[\text{D}_{\text{KL}}\left(\mathcal{E}\left[f_W(X)|W', X\right]\|f_W(X)\right)\right].
\end{aligned}
$$

Thus, here the misfit term $\mathbb{E}_{X,W,W'}\left[\text{D}_{\text{KL}}\left(\mathcal{E}\left[f_W(X)|W', X\right]\|f_W(X)\right)\right]$ matches the misfit term in Eq. (6).

In addition, if $f_{W'}(x) = \mathbb{E}\left[f_W(x)|W'\right]$ for $\forall x \in \mathcal{X}$, by Corollary 4.1 and Corollary 6.1, we have

$$
\begin{aligned}
\mathbb{E}_{X,W'}\left[\text{RCE}\left(g(X), f_{W'}(X)\right)\right] =& \mathbb{E}_{X,W}\left[\text{RCE}\left(g(X), f_W(X)\right)\right] - \mathbb{E}_{X,W',W}\left[\text{CE}\left(f_W(X), \mathbb{E}\left[f_W(X)|W', X\right]\right)\right] \\
& + \mathbb{E}_{X,W'}\left[H(\mathbb{E}\left[f_W(X)|W', X\right])\right] \\
=& \mathbb{E}_{X,W}\left[\text{RCE}\left(g(X), f_W(X)\right)\right] - \mathbb{E}_{X,W',W}\left[\text{D}_{\text{KL}}\left(f_W(X)\|\mathbb{E}\left[f_W(X)|W', X\right]\right)\right] \\
& - \mathbb{E}_{X,W}\left[H(f_W(X))\right] + \mathbb{E}_{X,W'}\left[H(\mathbb{E}\left[f_W(X)|W', X\right])\right].
\end{aligned}
$$

Since entropy is a concave function, we have $\mathbb{E}_{X,W'}\left[H(\mathbb{E}\left[f_W(X)|W', X\right])\right] \geq \mathbb{E}_{X,W}\left[H(f_W(X))\right]$. As a result, the performance gain between $\mathbb{E}_{X,W'}\left[\text{RCE}\left(g(X), f_{W'}(X)\right)\right]$ and $\mathbb{E}_{X,W}\left[\text{RCE}\left(g(X), f_W(X)\right)\right]$ is larger than that in Eq. (7).

This completes the proof. $\square$

### D.3. Proof of Proposition 2

**Proposition 2** (Restatement). *Given a data distribution $\mu = \mu_X \mu_{Y|X}$ and any $\alpha \in [0, 1]$, we define a label-shifted distribution $\hat{\mu} = \mu_X \mu_{\hat{Y}|X}$ by smoothing the labels as follows: for each $(X, Y) \sim \mu$, the smoothed label is given by $\hat{Y}_j = \frac{1}{2} + \alpha\left(Y_j - \frac{1}{2}\right)$ for $\forall j \in [2]$. If RCE is used as the loss function in binary classification, then the population risk minimizer on $\hat{\mu}$ also minimizes the population risk on $\mu$.*

*Proof.* Given any model $f : \mathcal{X} \to \mathcal{Y}$, define the population risk using reverse cross-entropy as

$$
R_{rce}(f) = -\mathbb{E}_X \sum_i [f(X)]_i \log Y_i,
$$

$$
R_{rce}^\alpha(f) = -\mathbb{E}_X \sum_i [f(X)]_i \log Y_i' = -\mathbb{E}_X \sum_i [f(X)]_i \log\left(\frac{1}{K} + \alpha\left(Y_i - \frac{1}{K}\right)\right),
$$

where supervision $Y = [Y_1, \cdots, Y_K]^T$ that satisfies $\|Y\|_1 = 1$. Therefore,

$$
\begin{aligned}
R_{rce}^\alpha(f) - R_{rce}(f) &= \mathbb{E}_X \sum_i [f(X)]_i \underbrace{\log \frac{Y_i}{\frac{1}{K} + \alpha\left(Y_i - \frac{1}{K}\right)}}_{q_\alpha(Y_i)} \\
&= \mathbb{E}_X \sum_i [f(X)]_i \, q_\alpha(Y_i). \tag{38}
\end{aligned}
$$

Note that $q_\alpha(Y_i) = \log \frac{Y_i}{\frac{1}{K} + \alpha\left(Y_i - \frac{1}{K}\right)} = \log \frac{1}{\alpha}\left(1 - \frac{1-\alpha}{1-\alpha+\alpha k Y_i}\right)$ is a monotonically increasing function of $Y_i$. Let the minimizer of the ambiguous weak supervision

$$
f_\star = \arg\min_f R_{rce}^\alpha(f).
$$

There always hold $R_{rce}^\alpha(f) \geq R_{rce}^\alpha(f_\star)$. Without loss of generality, let $Y_h = 1 - \epsilon$, where $\epsilon \to 0$. For the binary

classification case, i.e., $K = 2$, the inequality $R_{rce}^{\alpha}(f) \geq R_{rce}^{\alpha}(f_{\star})$ means

$$- \mathbb{E}_X \sum_i [f(X)]_i \log Y_i' \geq -\mathbb{E}_X \sum_i [f_{\star}(X)]_i \log Y_i'$$

$$\Rightarrow \mathbb{E}_X \sum_i [f_{\star}(X) - f(X)]_i \log Y_i' \geq 0$$

$$\Rightarrow \mathbb{E}_X [f_{\star}(X) - f(X)]_h \log Y_h' + \mathbb{E}_X [f_{\star}(X) - f(X)]_{1-h} \log Y_{1-h}' \geq 0$$

$$\Rightarrow \mathbb{E}_X [f_{\star}(X) - f(X)]_h \log Y_h' - \mathbb{E}_X [f_{\star}(X) - f(X)]_h \log(1 - Y_h') \geq 0$$

$$\Rightarrow \mathbb{E}_X [f_{\star}(X) - f(X)]_h \log \frac{Y_h'}{1 - Y_h'} \geq 0$$

$$\Rightarrow \mathbb{E}_X [f_{\star}(X) - f(X)]_h \geq 0$$

$$\Rightarrow \mathbb{E}_X [f_{\star}(X) - f(X)]_{1-h} \leq 0$$

By substituting $f_{\star}$ into Eq. (38) and we derive an additional similar equation. Combining this new equation with the original Eq. (38) yields the below results:

$$[R_{rce}^{\alpha}(f) - R_{rce}(f)] - [R_{rce}^{\alpha}(f_{\star}) - R_{rce}(f_{\star})]$$

$$= [R_{rce}^{\alpha}(f) - R_{rce}^{\alpha}(f_{\star})] - [R_{rce}(f) - R_{rce}(f_{\star})]$$

$$= \mathbb{E}_X \sum_i [f(X) - f_{\star}(X)]_i \, q_{\alpha}(Y_i)$$

$$= \mathbb{E}_X [f(X) - f_{\star}(X)]_h \, q_{\alpha}(Y_h) + \mathbb{E}_X [f(X) - f_{\star}(X)]_{1-h} \, q_{\alpha}(Y_{1-h})$$

$$= \mathbb{E}_X [f_{\star}(X) - f(X)]_{1-h} \, q_{\alpha}(Y_h) + \mathbb{E}_X [f(X) - f_{\star}(X)]_{1-h} \, q_{\alpha}(Y_{1-h}) \qquad (\|f_{\star}(X)\|_1 = 1)$$

$$= \mathbb{E}_X [f(X) - f_{\star}(X)]_{1-h} \, [q_{\alpha}(Y_{1-h}) - q_{\alpha}(Y_h)].$$

We have $Y_{1-h} \leq Y_h$, which means $q_{\alpha}(Y_{1-h}) \leq q_{\alpha}(Y_h)$. Also, $\mathbb{E}_X [f(X)]_{1-h} \geq \mathbb{E}_X [f_{\star}(X)]_h$. It leads to $[R_{rce}^{\alpha}(f) - R_{rce}^{\alpha}(f_{\star})] - [R_{rce}(f) - R_{rce}(f_{\star})] \leq 0$, i.e.,

$$0 \leq R_{rce}^{\alpha}(f) - R_{rce}^{\alpha}(f_{\star}) \leq R_{rce}(f) - R_{rce}(f_{\star}).$$

Thus, $R_{rce}(f_{\star}) \leq R_{rce}(f)$ holds for any $f$, which contributes to the final result

$$f_{\star} = \arg\min_f R_{rce}(f).$$

This completes the proof. $\qquad\qquad \square$

## D.4. Gradient Analysis

In the setting of predictive uncertainty, to further demonstrate the advantage of RCE described in Section 6, we introduce a gradient analysis for forward CE/KL and reverse CE/KL losses, which are shown in Eq. (39)-(42).

$$\frac{\partial \, \mathrm{CE}\,(g(X), f_{W'}(X))}{\partial \, [f_{W'}(X)]_j} = -\frac{\partial \, \sum_{i=1}^2 [g(X)]_i \log [f_{W'}(X)]_i}{\partial \, [f_{W'}(X)]_j} = -\frac{[g(X)]_j}{[f_{W'}(X)]_j} + \frac{1 - [g(X)]_j}{1 - [f_{W'}(X)]_j}, \tag{39}$$

$$\frac{\partial \, \mathrm{D}_{\mathrm{KL}}\,(g(X)\|f_{W'}(X))}{\partial \, [f_{W'}(X)]_j} = \frac{\partial \, \mathrm{CE}\,(g(X), f_{W'}(X)) - \partial \, H(g(X))}{\partial \, [f_{W'}(X)]_j} = \frac{\partial \, \mathrm{CE}\,(g(X), f_{W'}(X))}{\partial \, [f_{W'}(X)]_j}, \tag{40}$$

$$\frac{\partial \, \mathrm{RCE}\,(g(X), f_{W'}(X))}{\partial \, [f_{W'}(X)]_j} = -\frac{\partial \, \sum_{i=1}^2 [f_{W'}(X)]_i \log [g(X)]_i}{\partial \, [f_{W'}(X)]_j} = \log \frac{1 - [g(X)]_j}{[g(X)]_j}. \tag{41}$$

$$\frac{\partial \, \mathrm{D}_{\mathrm{KL}}\,(f_{W'}(X)\|g(X))}{\partial \, [f_{W'}(X)]_j} = \log \frac{1 - [g(X)]_j}{[g(X)]_j} - \log \frac{1 - [f_{W'}(X)]_j}{[f_{W'}(X)]_j} \tag{42}$$

As the student prediction $f_{W'}(x)$ approaches the supervision $g(x)$, the gradients for forward CE/KL and reverse KL diminish to zero. In contrast, the reverse CE gradient persists, maintaining non-vanishing values that facilitate more robust learning under uncertain supervision. It will be empirically validated in Appendix E.3.

# E. Experimental Details and Additional Results

## E.1. Experimental Details

**Dataset Processing**   For standard NLP tasks (SciQ (Welbl et al., 2017), Amazon Polarity (McAuley & Leskovec, 2013) and Twitter Sentiment), we convert the original questions and candidate answers $Q, A_1, ..., A_k$ into multiple question-answer pairs $(Q, A_i)$, where correct answers are labeled as positive and incorrect ones as negative. For reward modeling tasks (CAI-Harmless (Bai et al., 2022b), HH-RLHF (Bai et al., 2022a)), we directly pair the chosen and rejected responses while maintaining their original preference labels. All datasets maintain class balance and are partitioned into three subsets: $S$, $S'$, and $S_{test}$, designed for weak model training, strong model training, and final performance evaluation, respectively.

**Model Architecture**   For standard NLP tasks (SciQ, Amazon Polarity, Twitter Sentiment), with experimental setup of Burns et al. (2024), we add a two-dimensional linear projection layer atop the pretrained model's final representations, followed by a Softmax function to obtain the final prediction probabilities. For reward modeling tasks (CAI-Harmless, HH-RLHF), with experimental settings referencing Yao et al. (2025a;b), we instead use a single-output linear layer with a Sigmoid activation to map the output to a probability in [0, 1], indicating whether the response meets the harmlessness or helpfulness criteria. The added task-specific layer is randomly initialized before fine-tuning.

**Training Configurations**   We implement early stopping to prevent overfitting, using minimal training epochs for efficiency (Burns et al., 2024). Standard NLP tasks train for 2 epochs while reward modeling tasks use only 1 epoch. The base learning rate is set to $1 \times 10^{-5}$, except for GPT2 and GPT2-Medium models on SciQ and Amazon Polarity datasets which employ $5 \times 10^{-5}$. Batch sizes are configured as 32 for NLP tasks and 16 for reward modeling tasks. All experiments run on 4 NVIDIA vGPUs (32GB) with at least three independent random seeds for reproducibility. Complete configurations are summarized in Table 2. Notably, we adopt a full fine-tuning strategy during training without freezing any pretrained parameters, allowing the model to fully adapt to downstream tasks.

*Table 2.* Training Configurations on Different Datasets.

| Dataset | $|S|$ | $|S'|$ | $|S_{test}|$ | Max Seq Len | Epochs | Batch Size |
|---|---|---|---|---|---|---|
| SciQ | 5839 | 5838 | 1000 | 1024 | 2 | 32 |
| Amazon Polarity | 5000 | 5000 | 1000 | 1024 | 2 | 32 |
| Twitter Sentiment | 7000 | 7000 | 1000 | 1024 | 2 | 32 |
| CAI-Harmless | 4000 | 4000 | 4000 | 512 | 1 | 16 |
| HH-RLHF | 4000 | 4000 | 4000 | 512 | 1 | 16 |

**Loss Fuctions**   The Confidence-Adaptive Cross Entropy (CACE) loss dynamically combines reverse cross-entropy (RCE) and standard cross-entropy (CE) based on teacher confidence levels. Formally, it is defined as:

$$\mathcal{L}_{\text{CACE}}(y, \hat{y}) = \mathbb{I}(y, c) \cdot \text{RCE}(y, \hat{y}) + \big(1 - \mathbb{I}(y, c)\big) \cdot \text{CE}(y, \hat{y}), \tag{43}$$

where $\hat{y}$ represents the student's prediction and $\mathbb{I}(y, c)$ is an indicator function that activates when the teacher's soft label $y$ has confidence below threshold $c$. For our binary classification tasks, we quantify confidence as $|y - 0.5|$, where $y \in (0, 1)$ is the teacher's soft label. For multi-class classification tasks, where $y$ represents a probability distribution over $K$ classes, we measure confidence using the normalized Shannon entropy:

$$\text{Confidence}(y) = 1 - \frac{H(y)}{\log K},$$

where $H(y)$ denotes the Shannon entropy and $\log K$ corresponds to the maximum possible entropy (achieved by a uniform distribution), ensuring the confidence score remains within $[0, 1]$. The threshold $c$ is determined adaptively before W2S training by selecting a quantile such that exactly $\eta\%$, of the samples are viewed as low-confidence, i.e., activate $\mathbb{I}(y, c)$. In our experiments, $\eta \in \{5, 10, 20, 30\}$.

The Symmetric Cross Entropy Loss (SL) (Wang et al., 2019) is designed to handle noisy labels by balancing reverse cross-entropy (RCE) and standard cross-entropy (CE). It is formally defined as:

$$\mathcal{L}_{\text{SL}}(y, \hat{y}) = \lambda_1 \cdot \text{RCE}(y, \hat{y}) + \lambda_2 \cdot \text{CE}(y, \hat{y}), \tag{44}$$

where $\hat{y}$ represents the student's prediction, and $\lambda_1$ and $\lambda_2$ are weighting factors that adjust the contributions of RCE and CE respectively. In our experiments, we evaluate the performance of SL under different configurations of $(\lambda_1, \lambda_2) \in \{(1.0, 0.5), (1.0, 0.1), (1.0, 1.0), (0.5, 1.0), (0.1, 1.0)\}$.

The Auxiliary Confidence Loss (AUX) (Burns et al., 2024) is primarily aimed at enhancing the confidence of the student model's predictions through regularization. It is defined as:

$$\mathcal{L}_{\text{AUX}} = \beta \text{CE}(f_w(x), f_{w'}(x)) + (1 - \beta)\text{CE}(\hat{f}_{w'}(x), f_{w'}(x)), \tag{45}$$

where $\text{CE}(\cdot, \cdot)$ denotes the cross-entropy loss between two prediction distributions. Here, $f_w(x)$ represents the weak label prediction from one model, $f_{w'}(x)$ is the prediction from another weakly supervised model, and $\hat{f}_{w'}(x)$ is the hardened prediction of the student model, which becomes a one-hot vector when $f_{w'}(x)$ exceeds a certain confidence threshold. Following (Burns et al., 2024), the confidence threshold $t$ is adjusted so that exactly half of the samples in a batch have $f(x) > t$, ensuring that only sufficiently confident predictions are considered for hardening. Additionally, the weight parameter $\beta$ undergoes a linear warm-up phase, increasing from 0 to its maximum value $\beta_{\max}$ during the initial training period. For our experiments, we select $\beta_{\max} \in \{0.5, 0.75\}$ and apply this warm-up strategy separately over the first 20%, 50%, and 100% of the training data.

Notably, the results for CACE, SL, and AUX shown in Table 1 are the best outcomes selected from the above-mentioned hyperparameter settings.

### E.2. Bias and Variance Estimation in W2SG

Following (Yang et al., 2020), we provide the bias-variance decomposition for cross-entropy (CE) in our work, which is formally given by:

$$\underbrace{\mathbb{E}_X \mathbb{E}_\theta[\text{CE}\left(g(X), f_\theta(X)\right)]}_{\text{Population Risk}} = \underbrace{\mathbb{E}_X[\text{D}_{\text{KL}}\left(g(X)||\mathcal{E}_\theta[f_\theta(X)]\right)]}_{\text{Bias}^2} + \underbrace{\mathbb{E}_X \mathbb{E}_\theta[\text{D}_{\text{KL}}\left(\mathcal{E}_\theta[f_\theta(X)]||f_\theta(X)\right)]}_{\text{Variance}}, \tag{46}$$

where the model parameter $\theta \in \{W, W'\}$, and the intrinsic randomness of $\theta$ stems from several sources, such as data sampling, model initialization and optimization. Here, $X$ denotes a sample from the test set. $\mathcal{E}_\theta[f_\theta(X)]$ is the average of log-probability after normalization.

To accurately estimate $\mathcal{E}_\theta[f_\theta(X)]$ in Eq. (46), we train multiple independent teacher-student pairs on distinct subsets sampled from large-scale datasets and average their performance. Specifically, we construct five disjoint 10000-sample subsets from Amazon Polarity (with equal halves allocated to weak model training and W2S training), training five teacher-student pairs across four random seeds. Similarly, we extract three 14000-sample subsets from Twitter Sentiment, conducting experiments across five random seeds. The complete implementation details are provided in Algorithm 1.

To validate Corollary 4.1, which states that W2SG emerges when the student matches its (dual) posterior mean teacher, we implement a probability-based ensemble (Dietterich, 2000; Zhou, 2025) of multiple weak teachers for student supervision. Specifically, we compute the dual expectation $\mathcal{E}[f_W(X)]$ over weak teachers' predictions to approximate the teacher's conditional expectation $\mathcal{E}[f_W(X)|W', X]$ in Eq. (6). When using CE loss for teacher training, $\mathcal{E}[f_W(X)]$ is computed as:

$$\mathcal{E}[f_W(X)] = \frac{\exp\left(\mathbb{E}_W[\log f_W(X)]\right)}{\sum_{i=1}^{K} \exp\left(\mathbb{E}_W \log [f_W(X)]_i\right)}, \tag{47}$$

where $[f_W(X)]_i$ denotes the probability of sample $X$ belonging to class $i$ as predicted by the teacher model, and $K$ represents the total number of classes. Note that in this context, $X$ represents a fixed sample from the W2S training set, and the expectation is taken only over the weak teachers. We use these ensemble predictions as training data to supervise student models.

The results for the Amazon Polarity dataset are presented as dashed lines in Figure 1, and the results for the Twitter Sentiment dataset are shown in Figure 3. These results indicate that scaling up the student's capacity drives it closer to the posterior mean teacher (i.e., the black solid and dashed lines move closer in Figure 1). While this confirms that higher capacity facilitates the emergence of W2SG, we observe that the performance gain does not increase monotonically with model size. For instance, the CE loss of Qwen-7B is smaller than that of Qwen-14B in Figure 1. This suggests that an overly capable student may reduce the misfit term—the source of the gain—too aggressively, thereby limiting the magnitude of

---

**Algorithm 1** Estimate W2SG Bias and Variance

---

**Input:** Test point $\mathbf{x}$, weak model training set $S$ and strong model training set $S'$

**for** $i = 1$ **to** $k$ **do**

    Split the $(S, S')$ into $N$ pairs: $(S_1^{(i)}, S_1'^{(i)}), \ldots, (S_N^{(i)}, S_N'^{(i)})$

    **for** $j = 1$ **to** $N$ **do**

        Train the weak model $f_w$ using $S_j^{(i)}$

        Evaluate the weak model at $\mathbf{x}$; call the result $\pi_j^{(i)}$

        Generate pseudo-labels for $S_j'^{(i)}$ using $f_w$

        Train the strong model using $S_j'^{(i)}$

        Evaluate the W2S model at $\mathbf{x}$; call the result $\pi_j'^{(i)}$;

    **end for**

**end for**

Compute $\widehat{\pi} = \exp\left\{\frac{1}{N \cdot k} \sum_{ij} \log\left(\pi_j^{(i)}\right)\right\}$, $\widehat{\pi}' = \exp\left\{\frac{1}{N \cdot k} \sum_{ij} \log\left(\pi_j'^{(i)}\right)\right\}$

(*using element-wise log and exp; $\widehat{\pi}$ estimates $\bar{\pi}$*).

Normalize $\widehat{\pi}$ to get a probability distribution.

Compute the variance $\frac{1}{N \cdot k} \sum_{ij} D_{KL}\left(\widehat{\pi} || \pi_j^{(i)}\right)$, $\frac{1}{N \cdot k} \sum_{ij} D_{KL}\left(\widehat{\pi}' || \pi_j'^{(i)}\right)$.

Compute the bias $\frac{1}{N \cdot k} \sum_{ij} D_{KL}\left(\pi_0 || \widehat{\pi}\right)$, $\frac{1}{N \cdot k} \sum_{ij} D_{KL}\left(\pi_0 || \widehat{\pi}'\right)$.

where $\pi_0(\mathbf{x})$ be the one-hot encoding of the ground-truth label.

---

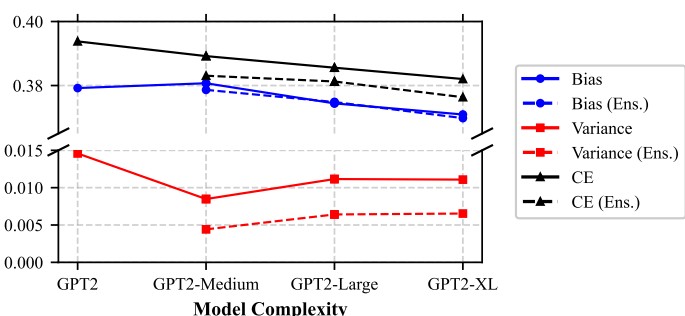

*Figure 3.* Decompose Cross-Entropy (CE) loss into bias and variance on Twitter Sentiment. Multiple weak teachers are employed to supervise student models. Solid lines represent the mean performance of students supervised by individual teachers, while dashed lines denote performance under teacher ensemble supervision (Ens.).

improvement. Importantly, This observation aligns with our theoretical analysis in Remark 5.1: while sufficient capacity serves as a condition for W2SG to occur (by ensuring convergence to the posterior mean), it does not necessarily maximize the achievable performance gain. Consequently, we acknowledge that there often exists an optimal student size that maximizes W2S performance, which is consistent with the empirical observations reported in Burns et al. (2024).

### E.3. Additional Empirical Validation of RCE's Robustness to Predictive Uncertainty

**Gradient Dynamics in CE and RCE** In addition to the Amazon Polarity and HH-RLHF datasets presented in Figure 2, we further compare the sensitivity of CE and RCE losses to predictive uncertainty on SciQ and CAI-Harmless datasets. To better understand why RCE outperforms CE under high uncertainty, we analyze their gradient dynamics First, we measure the extent of model updates by calculating the $L_2$ distance between the fine-tuned and initialized model parameters. As shown in Figure 2 and Figure 5, RCE consistently drives the model farther from its initialization compared to CE (i.e., the purple line is higher than the green line), except in the trivial case of $\alpha = 0$ where the RCE gradient vanishes. To explain this phenomenon, we track the gradient norm and the Gradient Direction Variance (GDV) (Liu et al., 2023), which quantifies

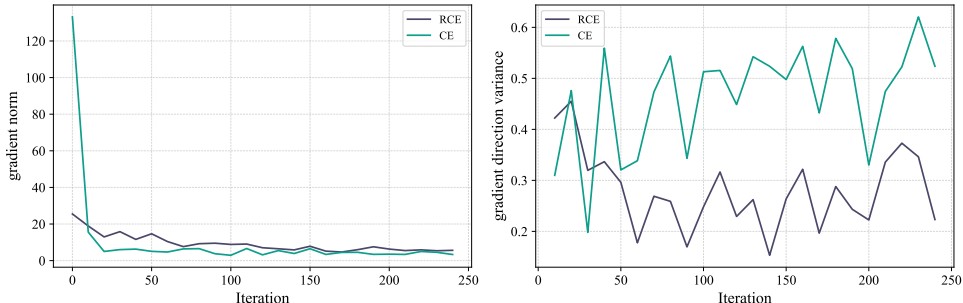

*Figure 4.* Gradient norm (left) and gradient direction variance (GDV, right) on the CAI-Harmless dataset, where GPT2-Medium is supervised by GPT2. RCE demonstrates lower GDV than CE, indicating stronger gradient consistency.

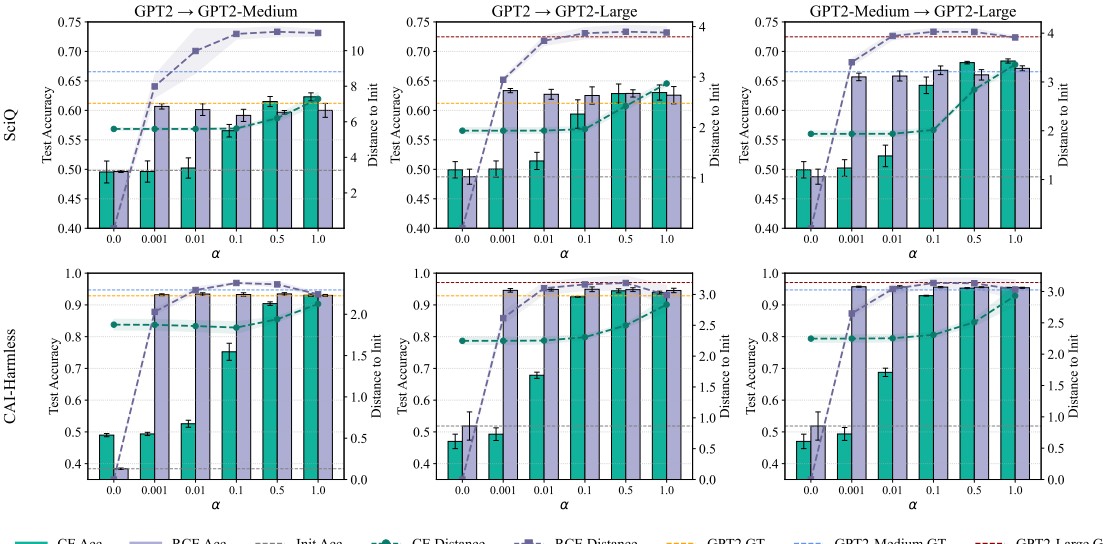

*Figure 5.* Comparison of CE and RCE losses across SciQ and CAI-Harmless. Lower $\alpha$ indicates higher uncertainty. "Weak $\rightarrow$ Strong" denotes the teacher-student pair. GT denotes training with ground truth labels. RCE demonstrates remarkable robustness to the teacher's prediction confidence. Left y-axis shows test accuracy, corresponding to the bar plots for CE Acc and RCE Acc. Right y-axis illustrates the $L_2$ norm between the fine-tuned and initial models, represented by the line plots for CE Distance and RCE Distance. RCE demonstrates remarkable robustness to the teacher's prediction confidence. All experiments are repeated three times.

quantifies the directional consistency of mini-batch gradients during training:

$$\text{GDV} = \frac{1}{|G| \cdot (|G| - 1)} \sum_{g_i, g_j \in G, i \neq j} \left( 1 - \frac{\langle g_i, g_j \rangle}{\|g_i\|_2 \cdot \|g_j\|_2} \right), \tag{48}$$

where $G$ denotes a set of mini-batch gradients. As shown in Figure 4, although RCE exhibits smaller gradient norms, its GDV remains significantly lower than that of CE. This indicates that CE updates are more "meandering" (high directional variance), causing the model to "wander" around the initial point. RCE compensates for smaller gradient magnitudes with high directional consistency, facilitating effective escape from the initialization region.

**Comparison of CE, RCE, KL and RKL in W2SG** Prior works (Yao et al., 2025a) investigated the mode-seeking behavior (Minka et al., 2005) of reverse KL divergence (RKL) in W2SG, establishing its advantages. We further conduct a systematic comparison of CE, RCE, KL, and RKL when applying the label smoothing strategy as outlined in Proposition 2. As shown in Fig. 6, RCE demonstrates significantly stronger robustness to predictive uncertainty compared to the other three losses on both CAI-Harmless and HH-RLHF datasets. This empirically validates Proposition 2 and Appendix D.4 regarding RCE's gradient stability advantages, despite RKL differing from RCE by only an entropy term.

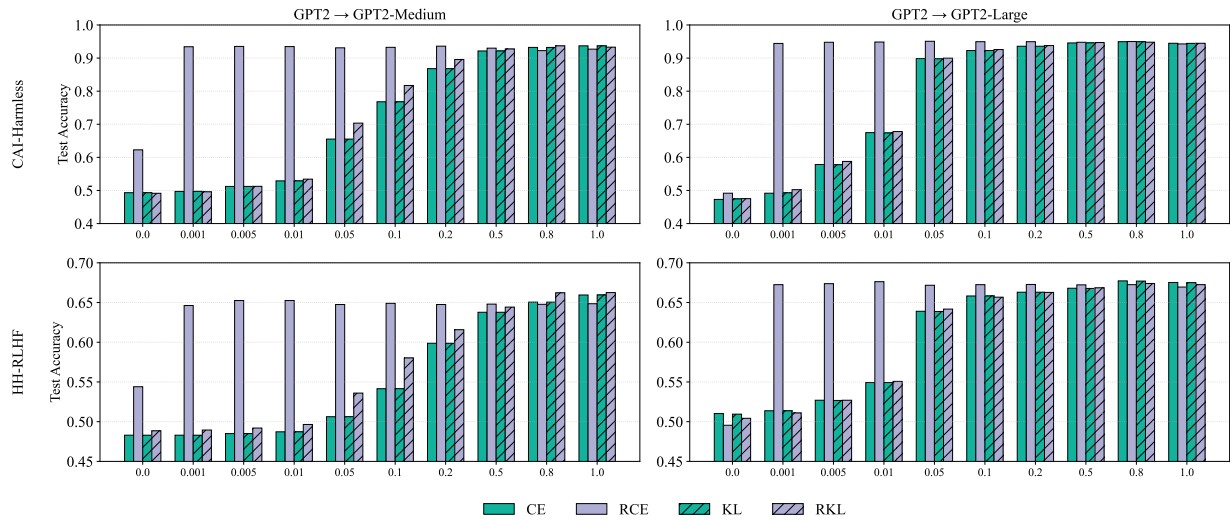

*Figure 6.* Comparison of CE, RCE, KL and RKL losses across CAI-Harmless and HH-RLHF. Lower $\alpha$ indicates higher uncertainty. "Weak $\rightarrow$ Strong" denotes the teacher-student pair. GT denotes training with ground truth labels. RCE demonstrates remarkable robustness to the teacher's prediction confidence.

**Comparison of CE and RCE in Knowledge Distillation**    We also investigate RCE under standard knowledge distillation (Hinton et al., 2015) settings, where a high-complexity teacher model supervises a low-complexity student model. Specifically, we consider two setups: (1) GPT2-Medium serves as the teacher, providing pseudo-labels to supervise GPT2; and (2) GPT2-Large supervises both GPT2 and GPT2-Medium. Results are summarized in Figure 7. Consistent with our findings in W2SG, RCE maintains significantly higher accuracy on samples with low-confidence predictions compared to CE. Models trained with RCE exhibit more consistent gradient directions and achieve larger parameter updates during training, indicating better learning stability and efficiency.

**Validation on Larger Models**    In addition to the GPT-2 series, we validated the comparative experiments of RCE and CE under the W2SG and Knowledge Distillation settings using the Qwen 2.5 series (Yang et al., 2024a) (Qwen-0.5B, Qwen-3B, Qwen-7B), as shown in Figure 8. Our results indicate that on more complex models, RCE continues to demonstrate its advantage in situations with low prediction confidence.

## F. Additional Experimental Results of CACE and SL

To verify the broad applicability of our approach, we extend our evaluation to include the Amazon Polarity dataset (text classification) and ImageNet (computer vision). As reported in Table 3, the results are consistent with our main findings: hybrid objectives (CACE and SL) consistently outperform other losses. This demonstrates that our proposed combination strategy is robust across diverse tasks and model architectures.

*Table 3.* Accuracy (%) of five loss functions on Amazon Polarity and ImageNet datasets. The optimal and suboptimal results are marked in **bold** and underline, respectively.

| Dataset | Model | CE | RCE | AUX | CACE | SL |
|---|---|---|---|---|---|---|
| Amazon Polarity | Qwen-1.8B $\rightarrow$ Qwen-7B | 95.50 | 96.30 | 95.80 | 96.50 | **96.60** |
| | Qwen-1.8B $\rightarrow$ Qwen-14B | 95.90 | 96.30 | 96.50 | **96.70** | 96.50 |
| | Qwen-7B $\rightarrow$ Qwen-14B | 96.40 | 96.40 | 96.90 | **97.10** | 96.80 |
| ImageNet | AlexNet $\rightarrow$ ResNet-50 (DINO) | 61.70 | 44.67 | 61.69 | 61.70 | **61.72** |
| | AlexNet $\rightarrow$ ViT-B/8 (DINO) | 66.68 | 54.63 | 68.06 | **70.03** | 68.42 |

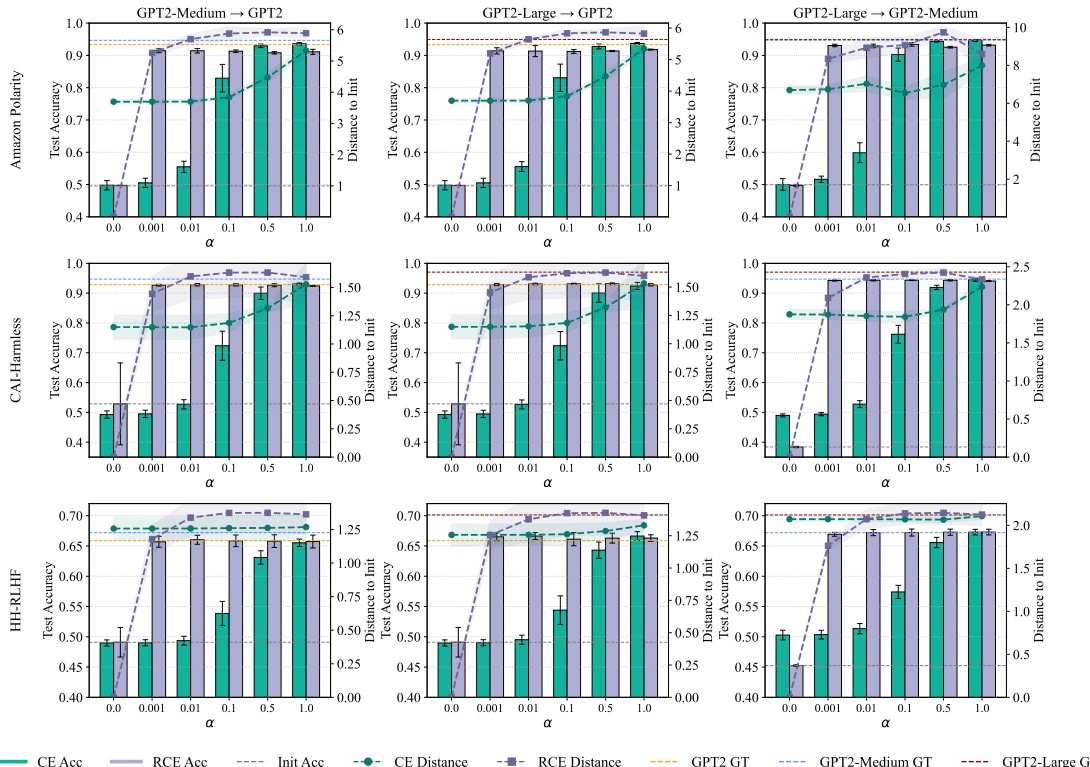

*Figure 7.* Comparison of CE and RCE losses across Amazon Polarity, CAI-Harmless and HH-RLHF under standard knowledge distillation. Lower $\alpha$ indicates higher uncertainty. "Strong $\rightarrow$ Weak" denotes the teacher-student pair. GT denotes training with ground truth labels. Left y-axis shows test accuracy, corresponding to the bar plots for CE Acc and RCE Acc. Right y-axis illustrates the $L_2$ norm between the fine-tuned and initial models, represented by the line plots for CE Distance and RCE Distance. RCE demonstrates remarkable robustness to the teacher's prediction confidence.

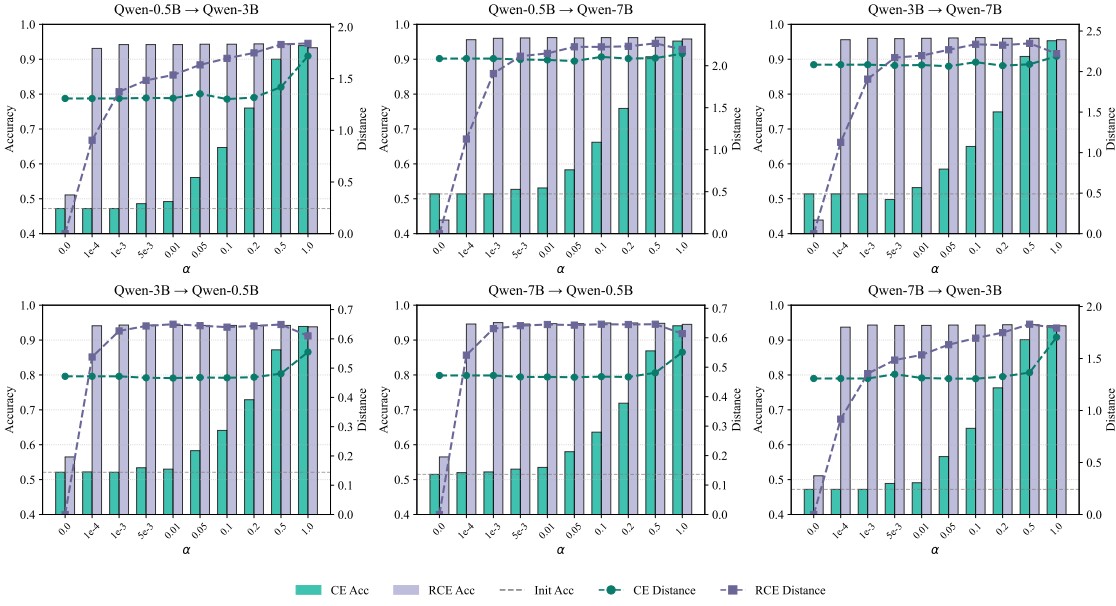

*Figure 8.* Performance of the Qwen series models on CAI-Harmless. Lower $\alpha$ indicates higher uncertainty. "Weak $\rightarrow$ Strong" denotes the teacher-student pair. GT denotes training with ground truth labels. Left y-axis shows test accuracy, corresponding to the bar plots for CE Acc and RCE Acc. Right y-axis illustrates the $L_2$ norm between the fine-tuned and initial models, represented by the line plots for CE Distance and RCE Distance. RCE demonstrates remarkable robustness to the teacher's prediction confidence.

