# OpenReview forum: "Weak-to-Strong Generalization via Bregman Bias–Variance Decomposition"
_ICML.cc/2026/Conference — ICML 2026 regular_

### Official Review · Reviewer_WEQK · 2026-03-01

**Soundness:** 3
**Presentation:** 3
**Significance:** 3
**Originality:** 3
**Overall Recommendation:** 5
**Confidence:** 3

**Summary:**

This paper develops a new misfit-based theory for weak-to-strong generalization (W2SG) grounded in a Bregman bias–variance decomposition. By working at the level of expected population risk and leveraging conditional/dual expectations, the authors remove convexity and realizability assumptions required by prior misfit analyses and obtain explicit residual terms that vanish when the student equals its (dual) posterior-mean teacher. Theoretical implications are instantiated for squared loss via an overparameterized ridge regression example that shows increasing student width drives convergence to the posterior mean with a nonzero misfit, and for cross-entropy via inequalities that highlight the role of student entropy and motivate reverse cross-entropy (RCE); empirical studies on GPT-2/Qwen families validate the main predictions and show practical benefits of RCE.

**Compliance With Llm Reviewing Policy:**

Affirmed.

**Final Justification:**

My concerns have been addressed after rebuttal.

**Key Questions For Authors:**

See weakness please.

**Limitations:**

yes

**Strengths And Weaknesses:**

- Strengths:
  - This paper Introduces a misfit-based W2SG analysis via Bregman bias–variance decomposition, replacing generalized Pythagorean arguments and thereby removing convexity/realizability assumptions.
  - It also provides new insight for asymmetric Bregman divergences (CE/KL), predicting the beneficial role of low-entropy student predictions and the robustness of RCE.
  - It empirically compares of CE vs RCE under controlled label uncertainty supports the theory, and additional confidence-adaptive losses show consistent gains across datasets and models.
  - The derivation and experiments are clear and comprehensive.

- Weaknesses:
  - Some notation and variables may be hard to parse for readers unfamiliar with dual expectations, and a brief intuition around duality could be explained.
  - I suppose the Qwen models are Qwen3 series? It is not mentioned in the paper, please make it specific.

---

> ### Author Rebuttal · Authors · 2026-03-31
>
> We sincerely thank the reviewer for the valuable feedback and for recognizing the theoretical and methodological contributions of our work. Our responses follow.
>
> > Some notation and variables may be hard to parse for readers unfamiliar with dual expectations, and a brief intuition around duality could be explained.
>
> We agree that the notion of dual expectation can be difficult to parse for readers unfamiliar with duality, and we will add a brief intuition before Definition 2.2 to improve readability.
>
> The core intuition stems from the asymmetry of Bregman divergences. In standard statistics, the expected value $\mathbb{E}[X]$ minimizes the "left-sided" Bregman divergence: $\arg\min _y \mathbb{E}[D _\phi(X, y)]$. Setting the derivative to zero directly yields $y = \mathbb{E}[X]$, which is why we can average points directly in the primal space.
>
> However, the dual expectation $\mathcal{E}[X]$ (Definition 2.2) minimizes the "right-sided" Bregman divergence: $\arg\min_y \mathbb{E}[D_\phi(y, X)]$. Taking the derivative with respect to $y$ and setting it to zero gives:
>
> $$
> \nabla\phi(y) - \mathbb{E}[\nabla\phi(X)] = 0
> $$
>
> This equation reveals why we cannot simply average the points $X$ to find the right-sided center $y$. Instead, we must first map points into the dual space via $\nabla\phi(X)$, average there, and then apply the inverse mapping to return to the primal space. Recalling that $\nabla\phi^*$ is the inverse of $\nabla\phi$ by properties of convex conjugation, this yields exactly our Definition 2.2:
>
> $$
> y = \nabla\phi^*(\mathbb{E}[\nabla\phi(X)]) \triangleq \mathcal{E}[X]
> $$
>
> For example, under KL divergence (cross-entropy), while the left-sided minimizer is the standard arithmetic mean, the right-sided minimizer yields a normalized geometric mean — which corresponds exactly to our ensembling procedure in Eq. (54).
>
> We believe adding this optimization-based derivation will make the concept of dual expectation transparent and natural to readers.
>
>
> > I suppose the Qwen models are Qwen3 series? It is not mentioned in the paper, please make it specific.
>
> We use the Qwen 2.5 series [1], with models up to 14B, to validate our theory and demonstrate the effectiveness of our approach. Thank you for pointing out this ambiguity, and we will revise the paper to make this explicit.
>
>
> [1] Yang et al., "Qwen2.5 Technical Report", arXiv.

---

> > ### Author Rebuttal · Reviewer_WEQK · 2026-04-01
> >
> > Thanks for response, and I will raise my score.

---

> > > ### Author Response · Authors · 2026-04-01
> > >
> > > We are pleased that our responses have addressed the reviewer's concerns and sincerely thank you for raising the score.
> > >
> > > Your suggestions — clarifying the intuition behind dual expectations and specifying the model series — have both led to a clearer presentation in the revision.
> > >
> > > Thank you again for the constructive and thoughtful review.

---

### Official Review · Reviewer_vEVs · 2026-03-04

**Soundness:** 3
**Presentation:** 2
**Significance:** 3
**Originality:** 3
**Overall Recommendation:** 4
**Confidence:** 4

**Summary:**

This paper studies w2s generalization and provide an insight for bias-variance decomposition by Bregman divergences. Specifically, the authors establish theoretical results to quantify the performance gap between the student model and the teacher model, showing that this gap is determined by the expected misfit between the two models. To validate their theoretical findings, the authors conduct empirical experiments using diverse datasets covering both standard NLP tasks and LLM reward modeling tasks.

**Compliance With Llm Reviewing Policy:**

Affirmed.

**Final Justification:**

My concerns have been adequately addressed

**Key Questions For Authors:**

1. My first question is about Example 1: the teacher and student models both appear to be linear models with an equal number of parameters, so how to define ''strong'' and ''weak'' here? Could you provide additional examples under more general model settings, such as nonlinear models?

2. I think the removal of the convexity assumption in Theorem 3.1 is a theoretical contribution, yet the example 1 still satisfies convexity. The authors should provide further discussion to illustrate the necessity of this theoretical contribution or supplement relevant examples to verify its validity in non-convex settings.

3. Could you provide more theoretical insights into w2sg in the presence of an ensemble of teachers? (This question will not affect my evaluation of the paper, and including a discussion on it would seem to improve the paper.)

**Limitations:**

Yes

**Strengths And Weaknesses:**

**Strengths**

1.  The experiments are comprehensive.

2. The theory is solid.

3. The authors provide a new perspective for studying w2s generalization by using Bregman divergences.

**Weakness**
While the removal of the convexity assumption in Theorem 3.1 is a theoretical advance, the paper fails to demonstrate the benefits of this relaxation.

---

> ### Author Rebuttal · Authors · 2026-03-31
>
> We sincerely thank the reviewer for the valuable feedback and for recognizing our contributions. Our responses follow.
>
> > W1 & Q2: Benefits of removing convexity; non-convex examples.
>
> Indeed, Example 1 satisfies convexity, as we acknowledge in the introduction (Lines 90–94). However, its main purpose is to show quantitatively that increasing student model size guarantees W2SG, thereby explaining why a larger student is typically needed in practice to observe this phenomenon. This result is entirely absent from previous misfit-based analyses, even where their convexity assumptions hold.
>
> We also clarify that, while an earlier version emphasized removing convexity itself, the current submission presents it as one aspect of our proof technique. The main novelty lies in the insights enabled by our framework: (i) identifying that approximating the posterior-mean teacher is a sufficient condition for W2SG, with an explicit characterization of performance gain; (ii) quantitative analysis of how student model size affects W2SG; and (iii) training implications via student confidence and CE/RCE losses.
>
> Nevertheless, while not our main claim, removing convexity does carry practical significance: it decouples the core mechanism of W2SG from structural constraints on the hypothesis class. This matters because modern deep learning models (with softmax) operate in non-convex hypothesis classes, as acknowledged in Mulgund & Pabbaraju (2025). In particular, the training implications in Section 5 require no convexity and can directly guide neural network training.
>
> Regarding non-convex examples, constructing rigorous analytical examples under non-convexity remains intractable given current double descent theory, we hope the reviewer appreciates that this difficulty reflects the current state of deep learning theory rather than a lack of interest on our part. However, our LLM experiments (GPT-2 and Qwen series) operate strictly in the non-convex regime and strongly align with our theoretical predictions, providing empirical verification that our framework governs non-convex deep neural networks. As theoretical tools for non-convex models continue to advance, our misfit-based framework is well-positioned for direct application.
>
> > Q1: Parameter counts in Example 1; nonlinear models.
>
> No, the two models do not have equal parameters. Following the original W2SG setup [1] we define "weak" and "strong" based on model capacity (i.e., parameter count).
>
> In Example 1, the weak teacher has $d_w$ parameters, while  the strong student has $(d_s \times d_w) + d_s$ parameters. Moreover, in Theorem 4.1, we consider the regime $\frac{d_s}{d_w} \to \gamma_1 \in (1, \infty)$, formalizing this capacity gap.
>
> Extending to nonlinear settings is certainly possible, but would require additional different mathematical tools, and such a generalization would likely constitute a separate research project in its own right. While Example 1 is presented using linear models for analytical clarity, similar phenomena have been established in more general non-linear regimes. For example, previous work on overparameterized random feature models [2] shows that increasing model size continues to reduce test error, consistent with our framework. This is therefore a natural direction for future work, though in our view it would be more appropriate to add these nonlinear extensions in a journal version of the current paper.
>
> Moreover, our empirical results (Figure 1, Lines 409–417) already demonstrate the effect of model size in non-linear settings, providing supporting evidence that our insights extend beyond linear models.
>
> [1] Burns et al., "Weak-to-strong generalization: Eliciting strong capabilities with weak supervision." ICML 2024.
>
> [2] Mei et al., "The Generalization Error of Random Features Regression: Precise Asymptotics and the Double Descent Curve." Communications on Pure and Applied Mathematics 2022.
>
> > Q3: Theoretical insights into ensemble of teachers.
>
> We would first like to clarify that model ensembling is not proposed in our paper as a new mechanism or algorithm. Rather, it serves as a practical proxy for the posterior-mean teacher to verify Theorem 3.1 and Corollary 3.1, i.e., primarily a sanity check.
>
> Theoretically, Section 3 shows W2SG arises when the student approximates its posterior-mean teacher; ensembling provides a natural way to approximate this teacher. Intuitively, this is also consistent with the classical *variance reduction* effect of ensembling, which obtains a more stable and informative supervision signal for the student.
>
> Moreover, Appendix B.2 further supports this by showing that improved performance can already emerge when the student approaches the expected teacher $\mathbb{E}_W[f_W(X)]$, which makes the usefulness of ensembles more transparent.
>
> We hope this clarifies the role of ensembles in our framework, and we will include a more explicit discussion of teacher ensembling in the revision.

---

> > ### Author Rebuttal · Reviewer_vEVs · 2026-04-02
> >
> > Thank you for the response. I have raised my score.

---

> > > ### Author Response · Authors · 2026-04-03
> > >
> > > Thank you for your thoughtful consideration and for taking the time to engage with our responses.
> > >
> > > We are pleased that your concerns have been adequately addressed, and we sincerely appreciate your consideration of adjusting the score.
> > >
> > > We will carefully revise the paper according to your suggestions.

---

### Official Review · Reviewer_Bgtf · 2026-03-09

**Soundness:** 2
**Presentation:** 2
**Significance:** 1
**Originality:** 1
**Overall Recommendation:** 4
**Confidence:** 3

**Summary:**

The paper studies weak-to-strong generalization (abbreviated W2SG) under Bregman losses, where the labels of a teacher model are used to train a student model that may (under some circumstances) outperform the teacher. It extends a previous work by Mulgund & Pabbaraju (2025) on W2SG by replacing a Bregman projection (and using generalized Pythagorean theorem) to prove the improvement by instead also considering the weights of the the models as random and studying a joint expectation. This yields sufficient conditions for W2SG in high-dimensional ridge regression, as well as classification with cross-entropy. Some experiments corroborate the findings.

**Compliance With Llm Reviewing Policy:**

Affirmed.

**Final Justification:**

The rebuttal addressed my main concerns regarding the "computability" of the posterior mean, relationship to ensembling, and a clear outline of the W2SG effect. The paper still seems somewhat limited in its contributions, but a weak accept seems adequate. Therefore, I raised my score.

**Key Questions For Authors:**

1. How does this work (and previous work) relate to literatures on ensemble learning and boosting?
2. Throughout the work, I am missing a section that clearly explains the effect why W2SG emerges here. Could you please do so?
3. In Tom Heskes (2025) „Bias-variance decompositions: The exclusive privilege of Bregman divergences“ an interesting addition to their considerations, it is shown that Bregman losses are essentially the only losses that enjoy a Bias-variance decomposition. Can this conclusion somehow be used to show that Weak-to-strong generalization only occurs for Bregman losses? Or is a Bias Variance decomposition only sufficient (and not necessary)?
4. In Wegel et al. (2025) „On the sample complexity of semi-supervised multi-objective learning“, their algorithm also fits one larger capacity model to the labels from smaller teacher models using Bregman losses. Could you explain how the mechanism with which the student outperforms the teachers there differs from the mechanism in you paper?

**Limitations:**

yes

**Strengths And Weaknesses:**

**Strengths**
The presentation of the work is mostly good and the writing is in a good state. The aim to generalize the projection argument from previous work seems sensible. The application in linear regression seems interesting (even though its not quite clear to me what the result is actually saying).

**Weaknesses**
In the previous work, Mulgund & Pabbaraju (2025), it is clear where the gain in „W2SG“ comes from: as the ground truth lies in a convex class, and the student is merely a projection of the teacher onto that class, the generalized Pythagorean identity guarantees the misfit gain. While that is arguably a very basic fact, almost tautological, it is clear what is happening.

In this work on the other hand, the main contribution is claimed to be that similar misfit gains can be shown without the convexity nor „realizability“ (that is, the ground-truth being contained in the convex hypothesis class). Instead the work relies on Bregman bias-variance decompositions. As i currently understand it, the W2SG effect in this work then appears as „denoising“: if the student is closer to the expectation („posterior mean“) of the teacher, then it reduces its variance. But of course this cannot be computed (e.g., the „estimator“ from Corollary 3.1 cannot be computed I think?). Indeed, only by using an ensemble of weak teachers, the posterior mean can be accessed (see also the experiments). But once there is access to multiple teachers, this is not surprising, and ignores a very long line of literature on ensemble learning and boosting.

So overall, I am somewhat confused what the point is.

**A small remark:**
I think the reference Bannerjee et al. (2005) „ On the optimality of conditional expectation as a Bregman predictor.“ should be cited, e.g., for Lemma 2.1. In particular, there are some (minor) technical assumptions necessary for that result.

---

> ### Author Rebuttal · Authors · 2026-03-31
>
> We sincerely thank the reviewer for the valuable feedback and recognition of our presentation. Our responses follow.
>
> > W1: Confusion about where the gain in W2SG comes from.
>
> > S1: Linear regression result unclear.
>
> > Q1: Relation to ensemble learning and boosting.
>
> Thank you for stating your main concern so clearly. We believe the confusion stems from interpreting Section 3 in isolation rather than viewing Sections 3–5 as a unified whole, which may make our paper seem to merely restate a previous W2SG inequality and motivate an ensemble-style training procedure—not our intent.
>
> Mulgund & Pabbaraju (2025) explain W2SG via a projection argument under realizability and convexity. Under these assumptions, the explanation of W2SG as arising from a projection argument is indeed straightforward and clear. However, we argue that such restrictive assumptions are unnecessary and, more importantly, do not provide a quantitative analysis of how student model size affects W2SG. In contrast, our work does not repeat the projection-based explanation, but aims to answer the question: *when these assumptions are relaxed, can misfit gain still be characterized, and why is W2SG more likely to emerge for larger students?*
>
> Concretely, Section 3 derives a W2SG inequality without restrictive assumptions and shows that approximating the posterior-mean teacher is sufficient for W2SG, but does not explain how this is achieved in practice. This is precisely the role of Section 4: we show that **increasing student capacity drives the student toward its posterior-mean teacher**, thus quantitatively explaining why a larger student is typically needed to observe W2SG, something not addressed in prior work. Section 5 then turns these insights into training implications via student confidence and CE/RCE losses. Thus, Section 3 is the theoretical foundation anchoring the full paper, not a standalone result.
>
> Regarding the posterior mean, we agree it is generally not computable. So we use model ensembling as a proxy to validate Theorem 3.1 and Corollary 3.1. While the form of our inequality naturally invites an association with ensembles, *the corresponding experiment should be interpreted as a sanity check of the theory, not an algorithmic contribution*. Moreover, our experiments are consistent with previous empirical observations (see introduction, Line 78), and model ensembling is not claimed as a contribution nor listed among the takeaways.
>
> We will expand the discussion of ensemble learning and boosting literature accordingly.
>
> > Q2: Clear explanation of why W2SG emerges.
>
> As summarized above: approaching the posterior-mean teacher is sufficient for W2SG (Section 3); concretely, increasing student size drives this convergence (Section 4), while higher student confidence further facilitates it (Section 5).
>
> We hope this clarifies why W2SG emerges in our framework and addresses the confusion raised in the weaknesses.
>
> > Q3: W2SG only for Bregman losses? (Tom Heskes, 2025)
>
> Thanks for your insightful question and reference.
>
> We note that Tom Heskes (2025) shows g-Bregman losses—broader than standard Bregman losses—are the only losses admitting a clean bias–variance decomposition. In this sense, the class considered in that work is already broader than ordinary Bregman losses.
>
> Clearly, we certainly do not claim that W2SG can occur only for Bregman losses. For example, in Section 5 we study cross-entropy (CE), which is not a Bregman loss but analyzable via KL divergence. We show both theoretically and empirically that W2SG can arise under CE given sufficient student confidence and capacity.
>
> > Q4: Difference from Wegel et al. (2025).
>
> In Wegel et al. (2025), teachers are trained on different tasks and the student learns a scalarized trade-off, not to outperform all teachers. By contrast, W2SG studies a student outperforming its teacher on one task. In our analysis especially Section 4, this stems from the student’s larger capacity enabling stronger representations and better generalization.
>
> Although our experiments also use multiple teachers, the goal is fundamentally different: ensembling serves as a proxy for the posterior-mean teacher on the same task, rather than balancing competing objectives across tasks.
>
> That said, both settings share a common structure: a larger-capacity model trained on pseudo-labels from smaller model under Bregman losses. Our bias–variance framework could potentially extend to multi-objective settings by analyzing per-task risk decompositions jointly. We view this as an interesting future direction and will discuss both Tom Heskes (2025) and Wegel et al. (2025) in the revision.
>
> > Remark: Cite Banerjee et al. (2005).
>
> Thank you for the suggestion. We will cite this foundational result showing conditional expectation is the unique optimal predictor iff the loss is a Bregman divergence.

---

> > ### Author Rebuttal · Reviewer_Bgtf · 2026-04-02
> >
> > Thank you for the clarifications. I think the presentation of the work requires significant changes to clarify which estimators are or are not computable, what the actual mechanism is that leads to W2SG here (and in particular, how it relates to ensembling and distilling multiple teachers into the student), and to discuss related works mentioned above.
> > As these changes are not so substantial to warrant a resubmission, i will consider raising my score.

---

> > > ### Author Response · Authors · 2026-04-03
> > >
> > > We sincerely thank the reviewer for the thoughtful follow-up and for being open to raising the score. We also greatly appreciate the suggestions on improving the presentation of our paper, and below we summarize how we will revise it.
> > >
> > > > Revisions to the Introduction.
> > >
> > > In the introduction, when discussing prior misfit-based works such as Mulgund & Pabbaraju (2025), we will explain more explicitly that their gain comes from a projection argument under convexity and realizability.
> > >
> > > Then, after presenting our Section 4 result on student model size, we will distinguish our source of gain from that of prior work, namely increasing student model size drives the student closer to the posterior-mean teacher, thus guaranteeing W2SG. This will help understand why W2SG is more likely to emerge for larger students.
> > >
> > > In addition, we will clarify that the posterior-mean teacher is a theoretical construct used to derive the key insights of our analysis, rather than a quantity that can be computed exactly in practice. We will also explicitly state in the introduction that *the teacher ensemble used in our experiments is only a proxy for verifiying Theorem 3.1 and Corollary 3.1, rather than an algorithmic advance*.
> > >
> > > Furthermore, we will also discuss potential directions for future extensions.
> > >
> > > > Computability of estimators (new Remark 3.1).
> > >
> > > After Corollary 3.1, we will add a new remark (Remark 3.1) clarifying that the posterior-mean teacher is generally not directly computable in practice. We will also make clear that the ensemble of weak teachers used in our experiments serves only as a finite-sample proxy for empirical validation, namely a sanity check rather than an algorithmic contribution, to avoid potential misunderstanding.
> > >
> > > More importantly, we will further explain that the sufficient condition involving the posterior-mean teacher is made concrete in Section 4 through student model size, so that this intermediate theoretical quantity leads naturally to the mechanism behind W2SG performance gain rather than remaining an isolated statement in Section 3.
> > >
> > > We believe this will improve coherence with the subsequent sections and make the underlying intuition more transparent.
> > >
> > > > W2SG mechanism (revised Remark 4.1).
> > >
> > > We will revise Remark 4.1 to give a more direct and intuitive explanation of why W2SG emerges in our setting, aligning it more closely with the new Remark 3.1 above. This flow can be summarized as
> > >
> > > $$
> > > \text{student model size} \uparrow \ \Rightarrow \text{convergence to posterior-mean teacher} \Rightarrow \varepsilon \to 0, \text{misfit} > 0 \Rightarrow \text{performance gain} > 0 \Rightarrow \text{W2SG occurs}
> > > $$
> > >
> > > We will also emphasize that, in this regime, simply increasing the student model size is sufficient, as the student naturally converges to the posterior-mean teacher, eliminating the need to rely on an ensemble as a proxy target for approximation.
> > >
> > > > Discussion of Related Works.
> > >
> > > We will incorporate the discussion of the references kindly pointed out by the reviewer.
> > >
> > > - Heskes (2025): In the Conclusion, we will discuss the theoretical connection to g-Bregman losses and clarify that W2SG is not limited to Bregman losses (e.g., our CE analysis in Section 5). We will further emphasize that, beyond empirical observations, extending our current theoretical framework beyond Bregman losses is an important direction.
> > > - Wegel et al. (2025): We will also compare our single-task W2SG setting with multi-objective learning (MOL) in the Conclusion, and discuss extending our Bregman bias–variance decomposition to MOL settings as a promising future direction.
> > > - Banerjee et al. (2005): We will cite this foundational result in the Preliminaries (Background on Bregman Divergences).
> > > - Ensemble learning and boosting: We will expand the discussion of related literature accordingly.
> > >
> > > We hope that these revisions will meet the reviewer's requests on presentation. Thank you again for the careful and constructive feedback.

---

### Official Review · Reviewer_B7Z3 · 2026-03-12

**Soundness:** 3
**Presentation:** 3
**Significance:** 4
**Originality:** 4
**Overall Recommendation:** 5
**Confidence:** 4

**Summary:**

- The paper provides a novel weak-to-strong generalization bound (W2SG) from the expected misfit lens. The bound relaxes the main assumptions used in prior work, which are the convexity of the hypothesis class and the realizability assumption.
- Furthermore, the bound provides a characterization of the residual error term epsilon.
- The paper provides analysis for a squared loss and cross-entropy loss (and reversed cross-entropy loss) and provides two key insights: i) enlarging the student model capacity leads to W2SG ii) aligning the student to the teacher via RCE is more desirable.
- Finally, the paper conducts experiments to validate these insights on GPT-2 and Qwen models.

**Compliance With Llm Reviewing Policy:**

Affirmed.

**Final Justification:**

We buttal have addressed my main concerns about the experiment setups.

**Key Questions For Authors:**

- How did you construct multiple weak teachers exactly for Figure 1?
- How did you set up the weak and strong model for the experiment in Figure 2? Also, why is the initial accuracy very low?
- How to pick the threshold c for CACE ?

**Limitations:**

yes

**Strengths And Weaknesses:**

**Soundness:**
- The theoretical result looks correct to me. The maths is rigorous, and the insights from the theorems are well explained.
- Experiment results are strong, especially in Figure 2, which shows that RCE is much more robust than CE, also in Figure 1, it is clear that ensembling leads to a lower variance, which leads to a smaller final CE loss. This suggests that the theoretical insights are practical.

**Presentation:**
- Introduction line 50-100 is quite dense. I found that it’s difficult to follow the results here.
- The notation and problem setup are clean and easy to follow.
- The details in the experiment section are not enough. I am a little bit confused with the experiment setup. In Figure 1, is it the CE loss of which student model ?  Is it always GPT2-XL and Qwen-14B ?
- In Figure 2, the dashed line is a little bit difficult to see. Further, while it is clear which model is weak and strong from “weak -> strong”, the training procedure is not very clear. It would be nice if the paper had a brief explanation of the setup in the main text.

**Significance:**
- The problem of weak to strong generalization is important
- The contribution is great, relaxing the convexity of the hypothesis class and the realizability assumption. Furthermore, I like that the theory here also provides insight for practical weak-to-strong fine-tuning.

**Originality:**
- The paper looks original to me

I am giving this current score because I am not sure if I fully understand the experiment setup. I am willing to change my score if the experiment setup is clearer.

---

> ### Author Rebuttal · Authors · 2026-03-31
>
> We thank the reviewer for the valuable feedback and recognizing our contributions, and greatly appreciate your openness to updating the score. We hope the detailed clarifications of our experimental setup below will provide a clearer understanding and resolve your confusion.
> > Introduction line 50-100 is quite dense. I found that it’s difficult to follow the results here.
>
> Line 50-100 is indeed quite dense, and we will revise to improve readability.
>
> For clarification, the main messages in Lines 50–100 are as follows:
>
> 1. We introduce new misfit-based W2SG inequalities and prove that a student model approximating its "posterior mean" teacher is a sufficient condition for W2SG, without restrictive assumptions.
> 2. In a ridge regression example, we show that increasing student capacity enables student model converge to its posterior mean teacher, facilitating the emergence of W2SG.
> 4. Through analysis of forward/reverse cross-entropy (CE/RCE), we demonstrate that high-confidence student predictions are more favorable for W2SG, and that RCE is better suited as the loss function under high teacher predictive uncertainty.
>
> These contributions represent the main theoretical advances in contrast to previous works, enabled by our generalized bias-variance decomposition of Bregman divergences.
> > The details in the experiment section are not enough. I am a little bit confused with the experiment setup. In Figure 1, is it the CE loss of which student model ? Is it always GPT2-XL and Qwen-14B ?
>
> Thank you for pointing this out.
>
> To clarify, the student model is not fixed to GPT2-XL or Qwen-14B in Figure 1. Instead, we fix the weak teacher for each model family and evaluate student models of increasing sizes. Specifically, we fix GPT2 as the weak teacher, with GPT2-Medium, GPT2-Large, GPT2-XL as the student models. We use Qwen-1.8B as the weak teacher, with Qwen-7B, Qwen-14B as the student models. For both teacher training and student W2S training, we use CE loss (the black line) and perform a bias-variance decomposition of the CE loss (the blue and red lines).
>
> We will add these setup details in the revised version.
> > In Figure 2, the dashed line is a little bit difficult to see. Further, while it is clear which model is weak and strong from "weak -> strong", the training procedure is not very clear. It would be nice if the paper had a brief explanation of the setup in the main text.
>
> Thank you agian for pointing this out, and we will revise Figure 2 to make the dashed line more visible.
>
> Additionally, we will include a description of the "weak -> strong" training pipeline in the main text to help readers better understand the experimental setup. Briefly, this involves two stages: (i) Teacher training: We train a weak model on data with ground-truth labels, using a standard supervised learning paradigm. (ii) W2S training: The weak model generates pseudo-labels for a new batch of data, which are then used to supervise a strong model. Other settings remain the same as in the teacher training stage.
> > How did you construct multiple weak teachers exactly for Figure 1?
>
> To construct multiple weak teachers, we follow [1] by training teachers on different data subsets. Specifically, we create multiple disjoint subsets (e.g., five 10k-sample splits from Amazon Polarity) and train separate teacher models on each subset (Line 1403).
>
> To combine them, we take the dual expectation (Eq. 54) over teacher predictions as a proxy for the ''posterior mean'' teacher in Eq. (6) (Corollary 3.1), which is then used to supervise the student:
> $$
> \mathcal{E}[f _{W}(X)] = \frac{\exp\left(\mathbb{E} _{W}[\log{f _W(X)}]\right)}{\sum _{i=1}^{K}\exp\left(\mathbb{E} _{W}\log{[f _W(X)] _i}\right)}
> $$
> We provide the full procedure in Algorithm 1 (Appendix E.2). We will add a brief summary of this construction process to the main text.
>
> [1] Yang et al., "Weak-to-strong reasoning", EMNLP 2024
> > How did you set up the weak and strong model for the experiment in Figure 2? Also, why is the initial accuracy very low?
>
> For the setup of the weak and strong models, as well as the training pipeline, please refer to our previous response. Regarding the low initial accuracy, we follow the standard W2S setup by adding a randomly initialized task-specific head on top of the pretrained backbone, which results in near-random initial predictions (around 50% accuracy). Additional implementation details are provided in the Appendix (Line 1305), and we would be happy to address any further questions on this.
> > How to pick the threshold c for CACE ?
>
> For the hyperparameter $c$, we follow the procedure described in Appendix E.1: the threshold is determined via a quantile-based selection of $\eta$, where $\eta$ specifies the percentage of samples treated as low-confidence. We evaluate several quantile levels $\eta \in \\{5, 10, 20, 30 \\}$ and report results using $\eta = 30$, which achieved the best performance on the validation sets of CAI-Harmless and HH-RLHF.

---

> > ### Author Rebuttal · Reviewer_B7Z3 · 2026-04-01
> >
> > Thank you for the rebuttals, the experiment setup is clearer and I have raised my score accordingly.

---

> > > ### Author Response · Authors · 2026-04-02
> > >
> > > We sincerely thank the reviewer for the positive acknowledgement and for raising the score.
> > >
> > > Your detailed questions on the experimental setup have greatly improved the clarity of our presentation, and the revised version will include explicit descriptions of the training pipeline, model configurations, and multi-teacher construction in the main text.
> > >
> > > Thank you again for the constructive feedback.

---

### Decision · Program_Chairs · 2026-04-30

**Decision:**

Accept (regular)

**Comment:**

**Summary:** This paper studies weak-to-strong generalization through the lens of Bregman bias–variance decomposition and proposes a new misfit-based analysis that removes convexity and realizability assumptions in prior work. Theoretical implications are instantiated for both squared loss and for cross-entropy, and empirical evaluations on GPT-2 and Qwen models validate the theoretical insights.

**Decision:** Overall, reviewers agreed that by leveraging Bregman bias–variance decomposition, this work made a solid theoretical contribution to the W2SG community. In particular, the reviewers noted that it offers a new perspective that removes the restrictive assumptions in existing works and yields practical training strategies.

Reviewers raised concerns about presentation and clarity, especially regarding experimental details, the gain of W2SG and its mechanism, the role of teacher ensembling, the motivation for the non-convex generalization, and notations related to dual expectations. The authors addressed these concerns well in the rebuttal by clarifying the weak-to-strong training pipeline, the multi-teacher construction, the mechanism behind W2SG, the role of ensembling as a proxy rather than an algorithmic contribution, as well as several presentation issues. These clarifications resolved the concerns, and all the reviewers raised their scores.

Given the theoretical contributions of this work and the consistency of the reviewers’ recommendations, I recommend acceptance.